# Intravital imaging by simultaneous label-free autofluorescence-multiharmonic microscopy

Sixian You[1,2], Haohua Tu[1], Eric J. Chaney[1], Yi Sun[1,3], Youbo Zhao[1], Andrew J. Bower[1,3], Yuan-Zhi Liu[1], Marina Marjanovic[1,2], Saurabh Sinha[4], Yang Pu[1] & Stephen A. Boppart [1,2,3,5]

Intravital microscopy (IVM) emerged and matured as a powerful tool for elucidating pathways in biological processes. Although label-free multiphoton IVM is attractive for its non-perturbative nature, its wide application has been hindered, mostly due to the limited contrast of each imaging modality and the challenge to integrate them. Here we introduce simultaneous label-free autofluorescence-multiharmonic (SLAM) microscopy, a single-excitation source nonlinear imaging platform that uses a custom-designed excitation window at 1110 nm and shaped ultrafast pulses at 10 MHz to enable fast (2-orders-of-magnitude improvement), simultaneous, and efficient acquisition of autofluorescence (FAD and NADH) and second/third harmonic generation from a wide array of cellular and extracellular components (e.g., tumor cells, immune cells, vesicles, and vessels) in living tissue using only 14 mW for extended time-lapse investigations. Our work demonstrates the versatility and efficiency of SLAM microscopy for tracking cellular events in vivo, and is a major enabling advance in label-free IVM.

[1] Beckman Institute for Advanced Science and Technology, University of Illinois at Urbana-Champaign, 405 N. Mathews Ave., Urbana, IL 61801, USA. [2] Department of Bioengineering, University of Illinois at Urbana-Champaign, 1304 W. Springfield Ave., Urbana, IL 61801, USA. [3] Department of Electrical and Computer Engineering, University of Illinois at Urbana-Champaign, 306 N. Wright St., Urbana, IL 61801, USA. [4] Department of Computer Science, University of Illinois at Urbana-Champaign, 201 N. Goodwin Ave., Urbana, IL 61801, USA. [5] Carle-Illinois College of Medicine, University of Illinois at Urbana-Champaign, 807 S. Wright St. Champaign, Urbana, IL 61820, USA. Correspondence and requests for materials should be addressed to H.T. (email: htu@illinois.edu) or to S.A.B. (email: boppart@illinois.edu)

Since the first demonstration in the late 1800s, intravital imaging has been transforming the way researchers probe and understand biology[1–3]. Instead of relying on static images to infer the possible interactions among different cell types, intravital imaging allows direct and longitudinal tracking of individual cells in their native environment. Enormous efforts have been devoted to developing molecular markers to be used for intravital imaging to visualize cells and structures of interest in vivo[2,4,5]. Although the growing library of markers (i.e., fluorescent molecules, nanoparticles, genetically expressed fluorescent proteins) has enabled biologists to scrutinize various components in the microenvironment, marker-based methods are fundamentally limited by the complicated and sometimes unpredictable tissue distribution of the exogenous markers, unexpected or unknown disturbance of biological or physiological functions, and unavoidable artifacts of non-specific labeling[6]. Label-free nonlinear optical microscopy, which produces high-resolution images with rich functional and structural information based on intrinsic molecular contrast, has demonstrated significant potential to overcome these problems by leveraging its non-perturbative nature and intrinsic molecular profiling capability. It eliminates the need to site-specifically label or target receptors or cells, and enables a broader array of volumetric signal generation from tissue structure and molecular composition.

Over the past two decades, a variety of biological phenomena have been investigated using images based on the autofluorescence of endogenous molecules excited by two/three-photon processes[7–11] or optical emission from specially structured molecules by second/third harmonic generation[7,12–14]. While these studies demonstrate excellent possibilities for label-free intravital imaging, each modality provides only a sub-set of the intrinsically accessible information, and promises to provide more comprehensive and informative images when augmented by complementary information from other modalities. A recent study further demonstrated the power of this approach for stain-free histopathology by integrating these four modalities, visualizing a variety of vital events in carcinogenesis including tumor cell migration, angiogenesis, and tumor-associate microvesicle enrichment[15]. However, similar to previous studies, two major drawbacks limited the application of this technology for further preclinical and clinical studies. First, the integration of these four contrast-generating processes usually requires different excitation bands and sequential image collection, which increases acquisition time as well as photodamage risk, and prevents rigorous spatial co-registration between sequentially detected signals, especially in living systems. Second, not all endogenous fluorophores and molecular structures in untreated biological tissues have large absorption cross sections, which directly leads to either long image acquisition time or poor image contrast.

Simultaneous implementation of multiple nonlinear imaging modalities has been demonstrated over the last decade. One type of platform simultaneously collected the structural information (noncentrosymmetry) of second-harmonic generation (SHG) and the functional information of two-photon autofluorescence (2PAF) for nicotinamide adenine dinucleotide (NADH, including its concentration and fluorescence lifetime)[16–19] and three-photon autofluorescence (3PAF) for serotonin excited at a short-wavelength (SW) band of ≤950 nm, and can thus be termed as SW-SHG&2PAF(&3PAF) imaging[7,9]. Another type of platform simultaneously collected the structural information of SHG and third-harmonic generation (THG, for optical heterogeneity) excited at a long-wavelength (LW) band of ≥1000 nm, and can thus be termed as LW-SHG&THG imaging[20,21]. These two platforms complement each other synergistically, but were problematic for efficient integration to simultaneously collect SHG, THG, and autofluorescence signals using one single (single-beam fixed-wavelength) excitation. The optimal spectral window for excitation and emission should also maximize the signal generation and detection efficiency for each multiphoton channel while ensuring orthogonality between them for spectral separation at detection. The need for THG contrast rules out SW platforms because of the strong UV absorption by tissue and standard optics. Therefore, there has long been a profound need to introduce functional information to the LW-SHG&THG platform in order to empower multiphoton microscopy as a versatile intravital imaging technique.

A major challenge with the development of a LW multiphoton platform is the significantly lower absorption cross sections of intrinsic fluorophores at longer excitation wavelengths. For example, our previous study showed high-quality 3PAF imaging of NADH with $1140 \pm 60$ nm excitation, but this was with an undesirable cost of imaging speed (pixel dwelling time ranged from 200 μs to 1 ms)[15]. In order to achieve real-time label-free imaging, signal generation efficiency had to be markedly improved without risking apparent photodamage. Prior studies have demonstrated fast 3PAF imaging by low-$f$ long-$\tau$ excitation (1 MHz, 509 fs, 1040 nm, 5.9 mW, and 1 μs per pixel)[22]. Since nonlinear optical signal (or photodamage) scales with $<I(t)>^n/(f\tau)^{n-1}$ ($n$—order of nonlinear process; $n = 2$ for SHG/2PAF and $n = 3$ for THG/3PAF)[23,24], it is expected that a combined low-$f$ and short-$\tau$ excitation condition, i.e., a large $(f\tau)^{-1}$ (inverse of duty cycle), would enhance the 3PAF signal at a given— $<I(t)>$, which should compensate for the decrease of multiphoton generation efficiency caused by the shift from SW to LW excitation[25]. The photodamage at a typical LW excitation (1080–1180 nm, 80 MHz, 100–250 fs, 120 mW, 3.3 μs per pixel)[26] indicates a nonlinear order $r$ between 2 and 3, just as in the case of SW excitation[27]. Thus, by the use of a larger $(f\tau)^{-1}$ coupled with a smaller $<I(t)>$, the third-order nonlinear imaging by 3PAF or THG gains a favorable signal-to-photodamage ratio (due to $3 > r$). Indeed, for a given imaging SNR, a short $\tau$ of 100 fs has mitigated the photodamage occurring at 250 fs in THG imaging[26].

Considering all these factors, we designed an optical imaging platform that performs simultaneous label-free autofluorescence-multiharmonic (SLAM) microscopy, featuring fast epi-detection of NADH from 3PAF and FAD from 2PAF, combined with noncentrosymmetric structures from SHG and interfacial features from THG (Fig. 1 and Supplementary Note 1). To eliminate the need for sequential excitation, we shift the excitation wavelength from the typical 740/900 nm band to a single-excitation band across 1080–1140 nm and simultaneously visualize various molecular contrast via four spectrally resolved detection channels. To overcome the challenge of weak intrinsic contrast, we use near-transform-limited excitation pulses with relatively broad bandwidth (60 nm, 35 fs) at a low pulse repetition rate (10 MHz), thus resulting in a significantly higher peak power compared to standard pulses (Supplementary Fig. 1). Images and real-time videos acquired from in vivo rat mammary tumors demonstrate the versatility and efficiency of SLAM microscopy for tracking intercellular and stromal–cell interactions in the non-perturbed (label-free) living tumor microenvironment. A vast array of cellular and stromal components, including tumor cells, vascular endothelial cells, blood cells, immune cells, and their surrounding extracellular matrix can be simultaneously visualized by SLAM microscopy with high spatiotemporal resolution. This work and novel method circumvents the aforementioned limitations in conventional intravital imaging and multiphoton microscopy by simultaneously and efficiently exciting autofluorescence and multiharmonic processes, and is a major enabling advance in label-free intravital imaging.

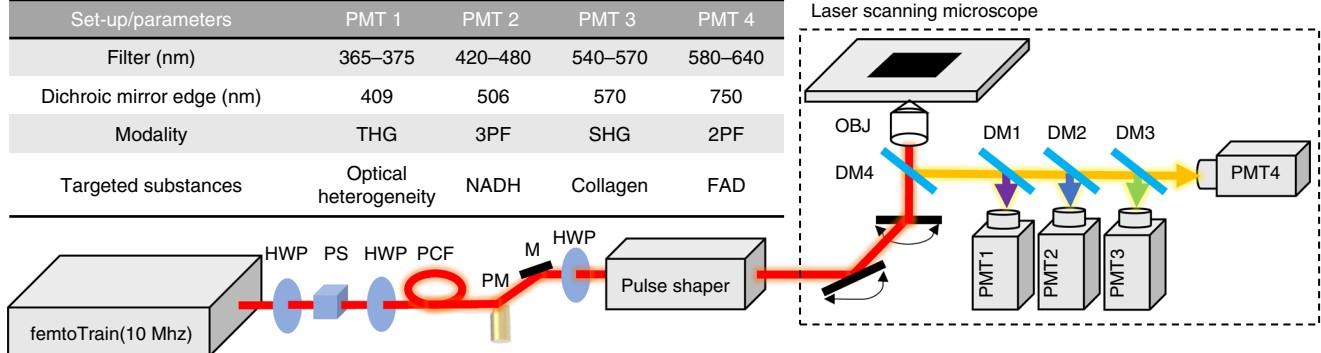

| Set-up/parameters | PMT 1 | PMT 2 | PMT 3 | PMT 4 |
|---|---|---|---|---|
| Filter (nm) | 365–375 | 420–480 | 540–570 | 580–640 |
| Dichroic mirror edge (nm) | 409 | 506 | 570 | 750 |
| Modality | THG | 3PF | SHG | 2PF |
| Targeted substances | Optical heterogeneity | NADH | Collagen | FAD |

**Fig. 1** Schematic of the SLAM microscopy platform. The high peak-power laser pulses were sent into the PCF to generate a supercontinuum. The pulse shaper is programmed to choose the excitation window (1080–1140 nm) and compensate the dispersion to make the output beam near-transform-limited. Different dichroic mirrors and optical filters are used in the detection system to collect spectrally resolved multimodal multiphoton signals by photomultipliers as specified in the table. DM dichroic mirror, HWP half wave plate, M mirror, OBJ objective, PCF photonic crystal fiber, PM parabolic mirror, PS polarizer splitter

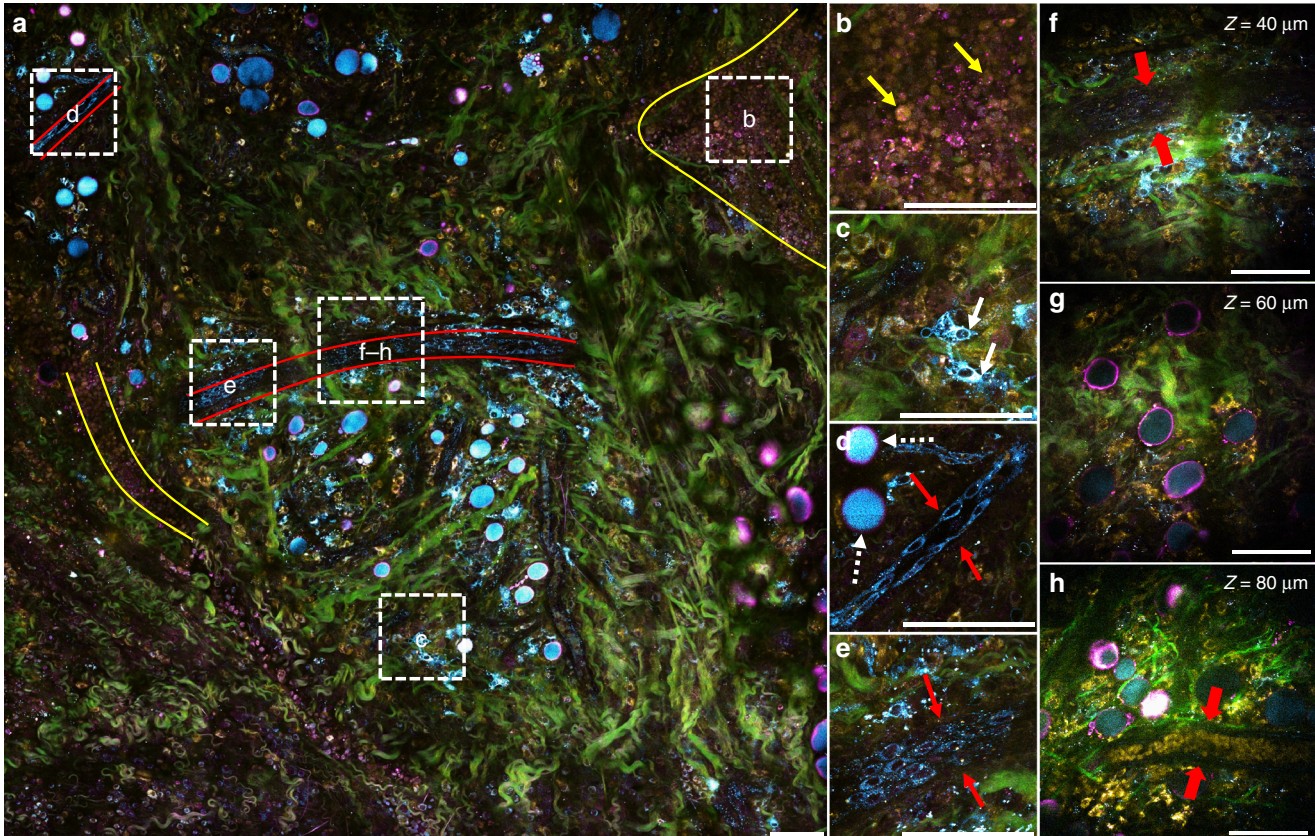

**Fig. 2** Tumor microenvironment of a living rat by SLAM microscopy. Pseudo-color presentation in the overlay image (consistent throughout this paper): green—SHG (collagen fibers); magenta—THG (interface); yellow—2PAF (FAD); cyan—3PAF (NADH). **a** Full view of a large field (1.5 × 1.5 mm²) at the tumor boundary, with different regions of interest (white dashed squares) magnified in **b–e**. Tumor cell clusters are highlighted with yellow boundaries and vasculatures with red boundaries. **b** 10-micron tumor cells (yellow arrows) with co-localized 2PAF (yellow, FAD) and THG (magenta, interface) signals. **c** Larger (>20-micron) cells (white arrows) with strong 3PAF (cyan, NADH) signals, which are likely to be macrophages due to their irregular cell shape and oval-shaped nucleus. **d**, **e** Vascular endothelial cells (red arrows) and adipocytes with THG-strong boundaries and NADH-strong content inside (white dashed arrows). **f–h** Images acquired from different depths at the center of **a**, with **f** displaying a hollow vessel composed of a layer of endothelial cells but no signs of red blood cells and **h** exhibiting a mature vessel filled with red blood cells. Contrast in **h** was intentionally brightened to show blood cells in deeper layers. Raw images can be found in Supplementary Fig. 4. Scale bar: 100 μm

## Results

**In vivo tumor microenvironment by SLAM microscopy.** Using an average power of only 14 mW at the sample surface, images 0.3 × 0.3 mm² in size were acquired sequentially (each within 3 s) and then mosaicked to form a larger field of view from an in vivo rat mammary tumor boundary. Various types of living cells were visualized within their natural environment, as shown in Fig. 2 and Supplementary Fig. 2, with a significant portion of them identifiable based on their distinct and reproducible features in morphology and 4-channel-based optical signatures. For instance,

tumor cells are identified as tightly packed round-shaped 10-micron cells with co-localized 2PAF (yellow, FAD) and THG (magenta, interface) signals, as shown in Fig. 2a and indicated by the yellow boundaries. The zoomed-in image (Fig. 2b) shows that tumor cells exhibited weak and homogenous FAD signals in their cytoplasm, mostly due to the increase of glycolysis over oxidative phosphorylation in (pre)cancer cells[28,29]. Punctate THG-associated diffraction-limited dots were found within the tumor, which are identified as tumor-associated vesicles based on prior studies[12,30]. The unique capability of SLAM microscopy for simultaneously co-localizing the 2PAF-associated tumor cells and THG-associated vesicles, together with the fact that these vesicles are usually washed out in conventional histopathology preparations (Supplementary Fig. 3), suggests the significant potential of combined THG and autofluorescence imaging for monitoring tumor-associated vesicular trafficking in vivo.

In addition to tumor cells, multiple subpopulations of stromal cells that play vital roles in carcinogenesis in the tumor microenvironment can be observed and followed. One important prognostic marker for cancer is the presence of inflammatory cells in tumors[31]. Figure 2c shows 3PAF-strong (cyan, NADH) cells (white arrows) with typical morphological features of macrophages, including irregular cell shapes, oval-shaped nuclei, cytoplasmic granules, and low nuclear-to-cytoplasmic ratios, as well as a significant increase in cell size[32]. Another cell type of interest are vascular endothelial cells, since angiogenesis is a hallmark of breast cancer progression[33]. Figure 2d shows two lines of 3PAF-strong (cyan, NADH) cells (white arrow) that are aligned and elongated along the same direction in a vascular structure, a morphology typical for vascular endothelial cells. Figure 2e shows another larger vessel layered with similar 3PAF-strong endothelial cells.

To have a more comprehensive view of the tumor microenvironment, we acquired volumetric images at the center of the same site (Fig. 2f–h, Supplementary Movie 1, and Supplementary Fig. 4). At a depth of 80 μm from the surface (Fig. 2h), the image revealed a vessel filled with 2PAF-associated (FAD, yellow) round biconcave disks with no signal void in the middle (no nucleus) as was observed in other cells, which are consistent with typical features of a perfused blood vessel containing red blood cells[34]. In contrast, the vessel (red arrows) in a shallower layer (Fig. 2f) was found to be comprised of 3PAF-associated endothelial cells with no evidence of luminal red blood cells, which resembles a hollow vessel. The co-existence of perfused vessels and hollow vessels within the same 3-D image stack validates published findings that aggressive tumors augment angiogenesis by forming hollow channels close to the existing vascular system[35]. Since microvessels and hollow channels are both frequently associated with poor prognosis in a wide range of human cancers[35], the capability of this system for visualizing microvessels and further distinguishing hollow vessels from mature vessels holds great promise for producing more selective prognostic biomarkers[36].

The color difference among the cells in Fig. 2 is likely to be a result of metabolic or structural differences. More interestingly, a ratiometric method using the 2PAF (FAD) and 3PAF (NADH) channels could potentially serve as an indicator of redox states for cells in vivo. As shown in Fig. 2, NADH signals (blue) are mostly associated with vascular endothelial cells while FAD signals (yellow) are mostly associated with tumor and stroma cells. This clear and consistent separation between the presumably NADH-rich cells and FAD-rich cells strongly suggest the feasibility of this approach for measuring redox state and further distinguishing not only their cell types but also their metabolic states. In addition, since every channel is acquired simultaneously under the excitation of a single beam from a single optical source, this approach is more robust in terms of motion artifacts and power fluctuations than conventional approaches where sequential (different) excitations are involved.

Consistent with previous studies[30], tumor-associated vesicles were abundantly seen near tumor boundaries and at sites of angiogenesis. Because of the significantly larger imaging FOV afforded by this platform, different subpopulations of vesicles can be seen in the same tumor microenvironment. More interestingly, as shown in Fig. 2b, THG-associated vesicles are usually associated with tumor clusters while 3PAF-associated vesicles are usually associated with vascular cells and structures, which suggests not only the existence of multiple subpopulations of vesicles in relation to different cell context within one tumor microenvironment, but also the ability to differentiate these subpopulations in vivo using these label-free optical signatures. More detailed investigations on the characterization of these seemingly different subpopulations of vesicles in association with different cell types are needed to understand this intriguing phenomenon. Nevertheless, combined with our recent findings[30], these new results further demonstrate the capability of SLAM microscopy to simultaneously visualize multiple subpopulations of vesicles together with various types of cells in living animals, which offers the potential to shed light on not only the cell-vesicle dynamics in carcinogenesis but also the clinically relevant changes that can serve as diagnostic and prognostic markers of cancer.

**Intravascular leukocyte recruitment.** Leukocyte recruitment from the circulation is a hallmark of the initial innate immune response and a key parameter in assessing immunotherapies. Using label-based intravital microscopy, leukocyte recruitment has been observed in different organs of living animals, and a series of sequential steps have been described: tethering, (slow) rolling, (firm) adhesion, crawling, and transmigration. Although it has been shown that the state-of-art label-based methods can identify different immune cell subsets as well as their surface phenotype and metabolic status[37,38], SLAM microscopy promises to provide complementary information on surrounding stromal and cellular components, which can be important to accurately identify molecular events or interpret the dynamic interactions. Here, using SLAM microscopy, we visualize the in vivo leukocyte recruitment cascade in the circulation without the use of any labels. Two modes of leukocyte arrest are featured in Fig. 3: slow rolling (Fig. 3a) and immediate arrest (Fig. 3b)[39]. As shown in Fig. 3a and Supplementary Movie 2, in the first few minutes, red blood cells are seen flowing rapidly in the vessels. Then a THG-associated multi-nucleated cell appears, and begins to roll along the vessel wall at a velocity of $1.06\,\mu m\,s^{-1}$, which is orders of magnitude slower than the perfusing red blood cells in the same vessel, displaying great resistance to the fluid drag force, and attributed to the strong attachment to E-selectin ligands on the vessel wall. These multi-nucleated cells are believed to be mature neutrophil granulocytes due to their clearly multi-lobed nucleus (Fig. 3a). The strong THG signals are in concert with the structural features of leukocytes (Supplementary Fig. 5 and Supplementary Note 3), which coherently enhance the THG process due to the abundant lipid bodies or vesicles within the leukocytes[12,14,40]. In particular, the lipid granules within neutrophils could further enhance the THG process, resulting in significantly higher THG contrast than in other leukocytes[40]. Figure 3b and Supplementary Movie 3 show a different site from another tumor-bearing animal, where we observed the other mode of leukocyte arrest (immediate arrest vs. slow rolling) in a vessel branch that borders a developing tumor cell cluster. The leukocyte in the time-lapse snapshots was observed to be captured and

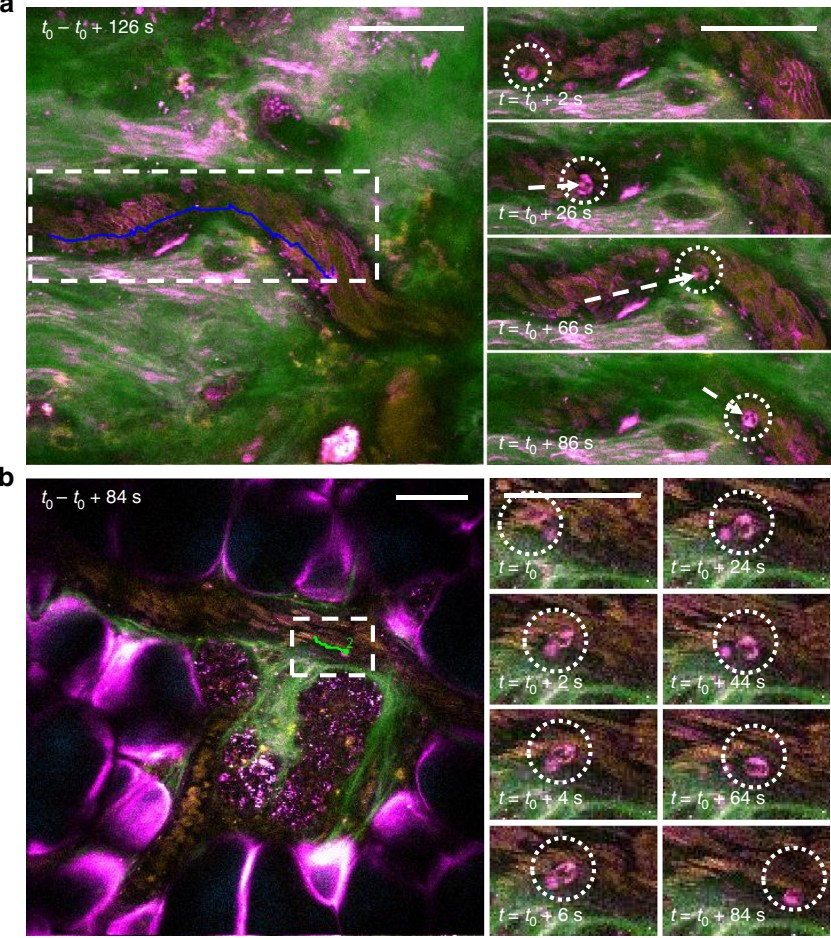

**Fig. 3** Two modes of leukocyte arrest captured by SLAM microscopy. **a** Multi-nucleated neutrophil slow rolling along the vessel wall at an instantaneous velocity of 1.06 μm s$^{-1}$. **b** Immediate arrest of a leukocyte with a sudden halt in the movement, followed by adhesion and crawling. Scale bar: 50 μm

tethered by the vessel wall with an abrupt deceleration in velocity, followed by a crawling motion, which typically precedes leukocytes extravasation[41]. Although it remains to be investigated whether the neighboring tumor cell cluster has orchestrated the immediate arrest of this leukocyte, these results demonstrate the capability of SLAM microsocpy to simultaneously visualize tumor cells, red blood cells, leukocytes, adipocytes, and collagen fibers in and surrounding the microvasculature with sufficient spatio-temporal resolution, and without any use of labels. Based on these results, there is a high potential for further noninvasive investigations of leukocyte extravasation in disease progression as well as in therapy assessment.

**Interstitial leukocyte migration**. A central motivation for developing this label-free imaging platform is to perform non-perturbative intravital imaging of intercellular dynamics and cell–stroma interactions. To examine the capability of SLAM microscopy for such tasks, and in addition to the intravascular motion tracking of leukocytes shown in Fig. 3, leukocyte loco-motion in the extravascular space was tracked with time-lapse imaging over a period of 70 min at one site that was 1 mm away from a spontaneously induced rat mammary tumor (Supplementary Fig. 6). As shown in Fig. 4a and b and Supplementary Movie 4 and 5, we observed marked interstitial chemotaxis of leukocytes that lasted ~50–70 min and resulted in the formation of several clusters at a site ranging from 30 to 70 μm in diameter. Although similar swarm-like behaviors of leukocytes have been

observed and described previously using multiphoton microscopy based on exogenous contrast[42,43], SLAM microscopy promises to create a more comprehensive picture and interpretation of this phenomenon owing to its unique capability to simultaneously excite autofluorescence and multiharmonic processes from the vast array of cellular and stromal components that comprise the tissue/tumor microenvironment. Consistent with previous experiments, multi-nucleated intracellular structures were observed for some of the THG-strong (magenta) cells (Fig. 4c–e), which resemble the morphology and optical signatures of neutrophils. In addition to the observed leukocyte recruitment and clustering, their interactions with the environment, including collagen fibers, adipocytes, and blood vessels, were simultaneously captured, putting the dynamic leukocyte behavior in context with the tumor/tissue microenvironment, and providing more insight into leukocyte behavior. Interestingly, since it was possible to simultaneously visualize collagen fibers using SHG and adipo-cytes using THG and 3PAF, the clearance of visible fibers and the deformation of lipid bodies was observed from the location where leukocyte clustering occurred, possibly through a highly orche-strated process. Longitudinal quantification of the visible fibers (SHG, green) and leukocytes (THG, magenta) within Cluster 1 (white box) showed that the density of the leukocyte cluster was steadily increasing while the collagen fibers began to disappear (either reorganized or degraded) only after the initial cluster had formed[43] (Fig. 4i). In parallel, the quantification of lipid content within Cluster 2 (cyan box) showed that over the time course of accumulation, the areas covered by lipids (NADH detected by

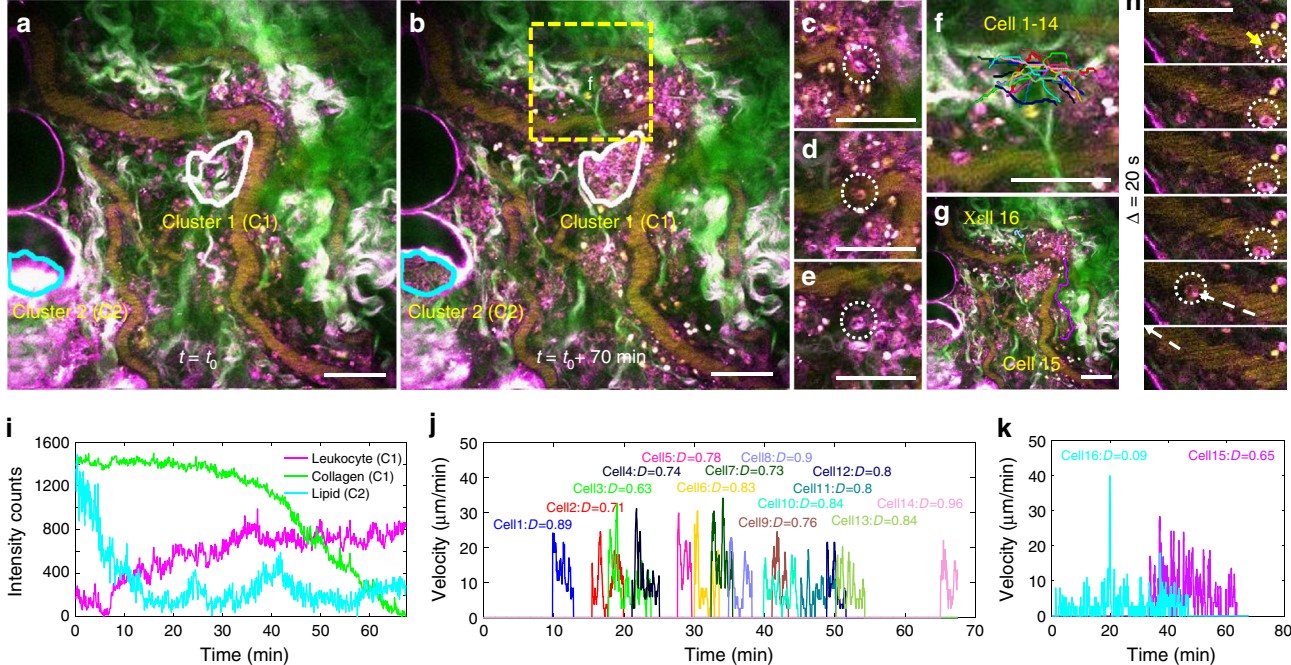

**Fig. 4** Leukocyte-swarming visualized and characterized by SLAM microscopy. **a**, **b** Images acquired at the beginning and the end of the swarming, respectively. Collagen rearrangement was marked by the white boundary (Cluster 1, C1) while lipid interaction was marked by cyan boundary (Cluster 2, C2). **c**–**e** Zoomed-in images of multi-nucleated neutrophils. **f** Traces of Cells 1–14, which were tracked to travel via similar routes to the same cluster at different time points, as shown in the velocity map **j**. **g** Traces of Cells 15–16, which were both tracked for at least 30 min and exhibited different behavior, with Cell 15 migrating towards the cluster with high speed and high directionality throughout the entire time course and Cell 11 mostly making random walk, as shown in the corresponding velocity map **k**. **h** Leukocytes leaving the site (Cells 17–18 in Supplementary Movie 5). The series of snapshots were taken every 2 s and shown for every 20 s. The first three snapshots showed the deformation of the leukocyte, changing from a round shape to a stretched cell elongated along the direction of travel, which typically precedes the acceleration process shown in the last three snapshots. **i** Quantification of collagen clearance, cell accumulation, and lipid deformation within the marked clusters in **a** and **b** (C1 and C2). **j**, **k** Velocity and directionality map of Cells 1–16. The color of the curves matches the color of the traces in **f** and **g** and Supplementary Movie 5. D directionality. Scale bar: 50 μm

3PAF, cyan) drop significantly, seemingly to clear the area for the leukocytes to congregate.

We next demonstrated that SLAM microscopy has sufficient spatiotemporal resolution to accurately and quantitatively describe cell locomotion on the single-cell level. Diverse migration patterns were observed in the same environment. A significant portion of leukocytes were recruited to the cluster with high directionality and at high speed, displaying peak velocities of more than 20 μm per min[43,44], as shown by Cell 1–15 in the velocity maps in Fig. 4j and k. Cell 1–14 could be attracted by short-range chemotactic signals due to their relatively short traces. Interestingly, despite the fact that these 14 cells show up at different time points throughout the 70 min, they share strikingly similar routes, speeds, and directionality when they are progressing to form the same cluster, which is in concert with the well-coordinated aspects of leukocyte-swarming behavior[43]. Cell 15 is representative of long-distance migration (~140 μm) with a similar high speed and directionality as Cell 1–14. Cell 16, however, was found to take a random walk path at a significantly slower speed for the entire time course, displaying a directionality that is five times smaller than those exhibiting highly coordinated chemotaxis (Cell 1–15). One commonly shared characteristic between the cells is their insect-like progression style, as documented and described previously for leukocytes and validated here[42]. As shown in Fig. 4j and k and Supplementary Movie 4 and 5, most traveling leukocytes moved forward via a series of repeated lunges, represented by periodic sharp peaks (rapid movement) in the velocity maps, followed by abrupt and distinct valleys (pauses).

Besides motion dynamics, SLAM microscopy also promises to provide metabolic profiling for individual cells at the same time. By comparing the redox ratio, FAD/(FAD + NADH), of clustering cells and migrating cells (Supplementary Fig. 7), a significant decrease ($p < 0.01$ by Student's $t$-test) in redox ratio was observed in clustered cells, indicating an increase in metabolic activity as well as a hypoxic environment[45]. This marked decrease of redox ratio in clustering cells is likely due to the combined effect of the large amount of energy consumed by cell transit and the diminished availability of oxygen at the center of cluster[46]. This preliminary analysis further demonstrates the power of SLAM microscopy to capture cellular dynamics and metabolic activities at the same time. To fully characterize and understand the metabolic changes in a triggered immune response, more in-depth experiments with neutrophils, macrophages, and lymphocytes are needed in future studies.

While the majority of extravasated leukocytes were likely directed by chemoattractants to the inflamed tissue and to form the cluster, several cells were observed to disappear together with blood flow. Opposite to the deceleration process shown in leukocyte arrest (Fig. 3), Fig. 4h showed the process of leukocyte acceleration in the vessel, with the leukocyte morphology beginning as a round cell (indicating a pause), becoming elongated along the direction of blood flow (indicating rapid movement), speeding up frame by frame, and finally disappearing into the stream of the rapidly perfusing red blood cells. Finally, during the time course of continuous imaging (70 min), no apparent laser damage was noted, and the dynamics observed by SLAM microscopy are largely consistent with previous reports of

leukocyte-swarming, confirming the non-perturbative nature of SLAM microscopy even after long-term intravital imaging.

## Discussion

We have demonstrated a single-excitation source nonlinear imaging platform that shows great promise for label-free, in vivo, high-resolution, 3-D, structural and functional imaging of cells and cellular dynamics in unperturbed complex environments. The advantage of SLAM microscopy over conventional intravital microscopy or multiphoton microscopy is that by using a custom-designed window of excitation wavelengths and shaped ultrafast pulses at a low 10 MHz repetition rate, it is possible to simultaneously and efficiently excite and detect autofluorescence and harmonic generation from a vast array of cellular and stromal components in living tissue. Moreover, long-term observation is possible because of the lower laser damage threshold afforded by the longer-wavelength excitation window as well as the low average power incident on the tissue. We anticipate SLAM microscopy to be an attractive alternative or complementary approach to existing intravital microscopy owing to its label-free nature, relative simplicity, versatility, and rich molecular profiling capability. Most importantly, few existing methodologies can provide a direct and fast multimodal visualization of unstained living tissue with such molecular, functional, and dynamic detail.

In addition, unlike marker-based intravital imaging approaches, the intensity of each channel in SLAM microscopy is proportionally correlated with the concentration of the autofluorophore or the arrangement of the molecular structures. The four-channel-based optical signatures can be quantified to objectively describe the metabolic and structural properties of individual cells, and thus identify different cellular and extracellular components as well as their functional states[30]. Although quantitative analysis of molecular structure based on SHG and THG intensity has been controversial due to their coherent nature[12,47], previous works have shown extensive analysis of the bulk optical parameters as well as morphological descriptions based on SHG and THG images[12,48]. The unambiguous separation between NADH-rich cells and FAD-rich cells indicates the strong potential of SLAM microscopy for in vivo redox state imaging in the future. Since 2PAF and 3PAF are simultaneously collected and expected to have identical incident power, we could potentially derive the relative concentration of NADH and FAD based on the known power dependence (power 2 for FAD and power 3 for NADH) and excitation efficiency of the system (calibration data from FAD and NADH solutions), which should provide a reliable qualitative assessment of the redox state. Furthermore, when compared to existing label-free nonlinear optical imaging techniques, SLAM microscopy promises to have more accurate multi-channel quantification owing to its single-source single-shot excitation and acquisition, which renders the analysis less susceptible to motion artifacts as well as power fluctuations, and eliminates the need to re-tune or re-align the laser and system optics between imaging channels.

The interaction between tumors, tumor cells, and individual components in the tumor microenvironment has become key to understanding how tumor cells proliferate, invade, disseminate and metastasize[35]. SLAM microscopy provides a powerful platform and method to probe the interactive dynamics between individual tumor cells and tumor-associated stromal cells in their native environment. In addition, different cell types are found to have unique and reproducible optical signatures, with 3PAF (NADH, cyan) mostly associated with endothelial cells and endothelial microparticles, 2PAF (FAD, yellow) found within almost all cells and only weakly present in tumor cells, and THG (interfaces, magenta) mostly appearing from tumor-associated

vesicles. Understanding why these optical signatures are associated with different cell types and even different cell functions and behaviors could shed light onto the complicated roles that different components have in cancer progression and could potentially contribute to more selective biomarkers for cancer diagnosis and prognosis.

Leukocytes have long been known for their fundamental roles in the early innate immune response and have recently been associated with key roles in the progression of metastatic cancer[49,50]. In contrast to current marker-based methodologies, SLAM microscopy provides a new and straightforward approach to uncover the spatiotemporal dynamics of leukocyte trafficking in living animal models with less perturbation but more morphological and molecular information from individual cells, subcellular features, and extracellular environment. We were able to track the major steps of leukocyte migration without aid of any markers, including recruitment from the circulation (tether, roll, adhesion, and crawling) and interstitial migration in the extravascular space (clustering and swarming), with high detail and in association with their interactions with the microvasculature, collagen fibers, and adipocytes, which we expect to greatly facilitate investigations and interpretations of leukocyte trafficking in not only the immune response but also in cancer metastasis. Leukocyte clustering is known to be triggered by various conditions of inflammation, including tumor progression, surgical operation, and laser damage. The possibility of this clustering being caused by the surgical procedure or laser damage is low, as the same experiments have been performed on control animals with no similar phenomena observed. Nevertheless, more controlled experiments are needed to pinpoint the exact trigger for this leukocyte cluster formation in future studies. It is to be noted that intravital microscopists have for decades utilized label-free transillumination in translucent tissues such as mesentery or cremaster muscle to study leukocyte traffic, and were able to differentiate leukocytes from surrounding cells such as endothelial cells and mast cells, thus revealing unique molecular events involved in the immune response[51,52]. The capability of SLAM microscopy to extend such intravital monitoring into 3-D opaque organs highlights the strong potential of SLAM microscopy to further deepen our understanding of leukocyte dynamics.

Nonetheless, we expect laser sources with other frequency shifting mechanisms, such as the Ti:Sapphire laser combined with an optical parametric oscillator (OPO), which has emerged as a powerful tool for multicolor label-based imaging[53], will also be suitable for SLAM microscopy on the condition that a strong excitation at 1110 nm is possible. The custom-designed fiber source is preferred for this work mainly because of its widely coherent spectrum (35 fs after compression versus 140–400 fs by OPO) and low repetition rate (10 MHz versus 80 MHz), which are essential for generating the high peak-power pulses that enable near-real-time recording of the intrinsically weak autofluorescence and multiharmonic generation signals (Supplementary Fig. 1). For future development of portable clinical systems, this source may also prove advantageous as the fiber-based system is simpler and more compact in design with lower-maintenance compared to the commercially available OPO system.

In summary, the presented SLAM microscopy platform and method achieves simultaneous visualization of a vast array of cellular and extracellular components in living tissue with high spatiotemporal resolution. We were able to image intercellular and stromal–cell interactions and dynamics over extended periods of time with only an average power of 14 mW on the sample. The label-free nature and the single-shot molecular profiling capability of SLAM microscopy could enable less perturbative and more comprehensive assessment of various biological,

physiological, and pathological processes in both basis research and clinical settings. Furthermore, the single-source single-excitation configuration of SLAM microscopy allows straightforward clinical translation to flexibly access challenging or deep tissue sites by an articulated arm, a miniature (laparoscopic) microscope objective, a fiber-based endoscope, or a handheld imaging probe, which may be further empowered by adaptive wavefront correction. For all these reasons, SLAM microscopy marks a transformative step for label-free intravital microscopy and a complementary addition to established marker-based technologies.

## Methods

**Fiber source**. The pump laser source for fiber supercontinuum generation was an extended cavity (10.2 MHz) industrial laser (femtoTrain, Spectra-Physics, Santa Clara, USA). The laser emitted 350-nJ 1040-nm 314-fs soliton pulses at an average power of 3.5 W. This high peak-power laser necessitated the use of a large mode-area fiber (PM-LMA-15, NKT Photonics, Denmark) to avoid photodamage to the fiber itself, and to enable high peak-power coherent supercontinuum generation[54,55]. This fiber was optimized over the dispersion-engineered small-core (2.3 μm) photonic crystal fiber that was previously intended for broadband coherent supercontinuum generation[15]. The generation of highly polarized coherent fiber supercontinuum and the construction of the corresponding pulse compression module largely followed our previous studies[30] (Supplementary Note 2, Supplementary Fig. 8 and Supplementary Table 1). The generated supercontinuum was sent into a 640-pixel 4-f pulse shaper (MIIPS Box640, BioPhotonic Solutions Inc.) with ~30% throughput. Only the 1110 ± 30 nm band of the supercontinuum was selected by amplitude shaping in the pulse shaper and used for subsequent SLAM microscopy. The average power of this band after the pulse shaper was ~50 mW.

**Optical setup for the SLAM microscopy system**. The SLAM microscopy system was implemented as an inverted multiphoton microscope (Fig. 1). The 1110 ± 30 nm pulses from the pulse shaper were raster scanned by a galvanometer mirror pair (6215 H, Cambridge Technology) and focused by a high UV-transmission objective (UAPON 40XW340, N.A. = 1.15, Olympus), with an average power of 14 mW incident on the sample after the loss along the excitation beam path. This combination of raster scanning and relatively low magnification microscope objective resulted in a typical field-of-view of $0.4 \times 0.4$ mm². The reflected multimodal multiphoton signals were spectrally-separated into 4 detection channels by long-pass dichroic mirrors and appropriate bandpass filters (Semrock, Inc.) and detected by 4 photon-counting photomultipliers (H7421-40, Hamamatsu) (Fig. 1 and Supplementary Note 1). The four emission filters were chosen to minimize the crosstalk between individual channels with specific bandwidth characteristics listed in the table inserted in Fig. 1. A live rat with surgically exposed mammary tissue was placed on a 3-D piezoelectric stage to enable mosaic-mode imaging and volumetric imaging, with the imaging plane placed 20–200 μm below the tissue surface. Pixel dwelling time ranged from 5–12 μs depending on the temporal and spatial resolution required for the application, typically resulting in a 2 s acquisition time for each image. Raw data from the photomultipliers were used to produce all images without additional processing such as deconvolution or maximum intensity projection. Because the only purpose of the pulse shaper was for spectral range selection and pulse compression of the excitation pulses, it may be replaced by a more common pulse compressor (e.g., a prism or grating pair) for further cost reduction.

**In vivo SLAM microscopy of a rat mammary tumor model**. Animal procedures were conducted in accordance with a protocol approved by the Institutional Animal Care and Use Committee at the University of Illinois at Urbana-Champaign. To induce mammary tumors in the female Wistar-Furth rats (Harlan, IN), NMU (N-Nitroso-N-methylurea) (Sigma, St. Louis, MO) diluted in distilled water (12.5 mg mL$^{-1}$) was injected intraperitoneally at a concentration of 55 mg per kg into the left side of the abdomen when the animals (Fisher 344, Harlan, Indianapolis, IN) were 7 weeks old. One week later, the same amount of NMU was injected intraperitoneally into the right side of the abdomen. After ~12 weeks of age, when mammary tumors became palpable, the rat was prepared for in vivo imaging. Surgery to expose the primary tumor and its neighboring mammary tissue was performed under isoflurane anesthesia. During the imaging sessions, each rat was anesthetized with 1% isoflurane mixed with O$_2$, at a flow rate of 1 L min$^{-1}$. Physiological temperature was maintained by a heating blanket. Imaging duration was kept under 3 h to avoid complications of long-term anesthesia. Rats were euthanized after imaging.

**Image processing and analysis**. In each frame, the four channels (THG, 3PAF, SHG, 2PAF) were acquired simultaneously and saved as individual four-channel images. For mosaic acquisition, tiles were stitched together using a custom-written MATLAB (Mathworks, Natick, Mass) script that matched sub-image boundaries

based on the predicted position map of each tile. For volumetric imaging and real-time imaging, the tiles were stacked and displayed as a four-channel video. The images shown in Fig. 4a–g were smoothed by averaging the neighboring 3 frames in the real-time video. The raw video and the video after smoothing are shown in Supplementary Movie 4 and 5, respectively. Without any further preprocessing, the raw four-channel image/video was loaded onto FIJI (National Institutes of Health) to apply pseudo-color maps to merge the contrast. The same color maps were used consistently throughout the study, namely magenta (hot) for THG, cyan (hot) for 3PAF, green for SHG, and yellow (hot) for 2PAF. This specific color scheme was chosen mainly for two reasons: (1) it improves the contrast between the cellular and extracellular components of interest; and (2) it provides intuitive interpretation of the mechanism of the contrast as each individual channel is assigned with a color that approximates the true emission wavelength.

The measurement of areas covered by leukocytes in Fig. 4i was conducted based on the prior knowledge that leukocytes were the major source of THG signals in the specified cluster throughout the measurement window. The same principle applied to SHG signals representing collagen fibers and 3PAF for lipids in the specified cell cluster. Thus, by counting the number of pixels with strong THG, SHG, or 3PAF signals (determined by universal thresholding), a rough estimation of the density of leukocytes, collagen fibers, or lipids could be achieved. The individual cell trajectories were extracted from the real-time videos using FIJI (Plugins, Manual tracking). Then, the trajectories were analyzed by a customized MATLAB script to quantify the speed and directionality of individual cells. More specifically, the instantaneous velocity was computed as the migrated distance over 18 s, while the directionality was defined as the displacement divided by the total path length of the cell[56].

**Code availability**. The codes that are used to generate the quantitative results in this manuscript (e.g. analysis of cell trajectories) are available from the corresponding author upon reasonable request and through collaborative investigations.

**Data availability**. The data that support the findings of this study are available from the corresponding author upon reasonable request and through collaborative investigations.

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

## Acknowledgements

We thank Newport (Spectra-Physics) and Biophotonic Solutions for their material support in laser source development. We also thank Darold Spillman for his technical support. This work was supported in part by grants from the National Institutes of Health (R01 CA166309, R01 CA213149, R01 EB023232, S.A.B.). S. Y. acknowledges support from the McGinnis Medical Innovation Graduate Fellowship and the Computational Science and Engineering Fellowship. Additional information can be found at http://biophotonics.illinois.edu.

## Author contributions

S.Y., H.T., and S.A.B. conceived the idea of SLAM microscopy. H.T. developed the laser source based on nonlinear fiber optics. S.Y., Y.S., Y.Z., A.B., and Y-z.L. designed, built, and programmed the optical imaging system. E.J.C., M.M., and S.A.B. planned the animal experiments. E.J.C. and M.M. performed the animal procedures. S.Y. conducted the imaging experiments. S.Y., H.T., S.S., Y.P., and S.A.B. analyzed the data and wrote the manuscript with input from all authors. S.A.B. and H.T. obtained funding and supervised the research.

## Additional information

**Competing interests:** The authors declare no competing interests.

