## [Peer Review File · Nature Communications]

Reviewers' comments:

Reviewer #1 (Remarks to the Author):

This is a very interesting study demonstrating a label-free approach to generating in vivo images differentiating multiple cell types in vivo using non-linear methods and a single excitation beam, and a simultaneous multichannel approach. It shows the capacity to differentiate between normal stroma and tumour cells, the vasculature and immune cells all in the absence of exogenous staining or endogenously-expressed reporter proteins. From an immune cell perspective, the videos convincingly show the capability of the system to detect slow rolling and crawling in blood vessels, and impressively even extravascular leukocyte migration in the tumour-associated tissue, in a label-free system.

Comment:

A devil's advocate view would be that intravital microscopists for many years used transillumination in tissues such as the mesentery or cremaster muscle to study leukocyte dynamics in and out of vessels without using staining/reporter proteins, and were able to differentiate leukocytes, endothelial cells, mast cells etc. This work provides an extension of this by doing it in 3 dimensions in opaque organs, which is clearly an important advance and should be emphasised.

However the authors should comment on the ability of this system to detect different types of leukocytes. The capacity of the SLAM system to detect neutrophil granulocytes is stated to be due to the abundant lipid bodies these cells possess. However, other important immune cells, particularly T cells, carry few lipid granules. How well does this system go in detecting these? Interestingly in Video 3, other leukocytes can be seen, as negative/unstained shapes adhering in the blood vessel, indicating that whatever type of leukocyte this is, it is not being (positively) detected by this system, and presumably would not be visible outside the vasculature. Are these also granulocytes, or might they be other cell types? It would be interesting to examine for example a lymph node with this system in that this organ is full of lymphocytes and therefore would give an idea of the capacity of this approach to detect other types of immune cells. If it is less effective for these cells, that would be useful information.

Also, would it be possible to confirm some of the observations in regards to immune cell identity by performing similar experiments using established approaches making use of labelling to confirm the identity of the immune cells examined (this may not be straightforward in a rat-based system)?

Minor comments:

Intravital microscopy has been in use for over 100 years, and gained prominence in the 1980s, so the opening sentence should be modified.

The statement on p8 about the limited capabilities of current methods to provide morphological/molecular information on different types of immune cells is not a true representation of the current state of the field – it is possible now to definitively identify different immune cell subsets, even rare ones like regulatory T cells, as well as studying their surface phenotype (e.g. adhesion molecule expression) and metabolic state (e.g. ROS production). This sentence should be rewritten to reflect this.

There are some technical details I think it would be beneficial to add:

- Depth of imaging. It is stated that images were taken from 20-200 um below the surface. It would be useful for assessing the utility of this approach if example images could be provided at different defined depths, along the lines of the first video (out to 115 um). Providing a visual demonstration of where these signals start to drop out, and how each of the types of signal is individually affected would be useful.

- Time resolution – how quickly can these 3D images be acquired?
- What are the bandwidth characteristics of the filters splitting the emitted signal into the respective PMTs?

Reviewer #2 (Remarks to the Author):

The manuscript of You et al presents simultaneous label-free auto-fluorescence microscopy (SLAM) and its use for intravital imaging in rodents. The innovative approach adapts and applies the excitation source, the authors developed and published in the context of histopathology, to the demanding field of intravital microscopy. The power of the technology is demonstrated in tumor environment context – a field of highest scientific impact – showing its advantage regarding the acquisition speed as well as the possibility to monitor tissue and cellular metabolic function in vivo. I expect that the here presented technology will open unprecedented perspectives for the label-free intravital functional imaging for various branches of biomedical research, not being limited to the here discussed application.

While the entire experimental work has been carefully performed and the results are well interpreted, in order to make the manuscript thoroughly valid, the SLAM technology needs to be better placed in the context of already existing intravital imaging technologies (both label-free or labeling-based):

(The authors discuss the possibility to perform single wavelength excitation at a time in a multi-photon microscope and correctly refer to the work from the Debarbieux lab regarding sequential single-wavelength excitation multiplexed in vivo imaging. In this context, the four-color label-free intravital imaging after excitation at 1110 nm (10 MHz) is the unique way to perform simultaneous imaging of NAD(P)H, FAD next to SHG and THG. However, in the manuscript, the fact that simultaneous multiple wavelengths excitation in a multi-photon microscope has been successfully performed and is now an established intravital imaging technique is completely omitted (Herz et al, *Biophys. J.*, 2010; Mahou et al, *Nat. Methods*, 2012; Rakhymzhan et al, *Sci. Rep.*, 2017). For a comprehensive understanding, the authors should comment in more detail on the possibility of using a combined simultaneous excitation at 740 nm and 1110 nm (nowadays available in standard multi-photon microscopes, from Ti:Sa lasers combined with optical parametric oscillators or amplifiers) instead of 1110 nm alone using their custom-made excitation source. An argument for their excitation source is surely the lower repetition rate of 10 MHz together with the relatively narrow pulse width– which allow for a more effective excitation, better photon management and, thus, higher acquisition speeds and, eventually, lower photobleaching and photodamage. Still, this advantage has not been demonstrated. At least a comment on this issue and on the possibility to use well-established, commercially available laser amplification (pulse picker / Pockels cells etc.) is necessary, beyond the already cited solutions applied in exclusively label-free IVM.

Hence, the following questions arise: (1) Why is a custom-made, not easily available single-wavelength excitation better than multiple-wavelength excitation, available in standard multi-photon microscopes? (2) Why is the custom-made excitation source better than the “old” solutions for the “old” requirement of reducing the laser repetition rate and increasing pulse energy and peak power (i.e. narrow pulse width) to achieve a higher signal at low mean laser powers.

(The authors point out at the possibility to use their technology to quantify cellular metabolism and to differentiate between oxidative phosphorylation and glycolysis – a reliable marker to distinguish between highly active tumor cells and normal/healthy cells in tissue. The possibility to monitor tissue and cell function is invaluable and is greatly acknowledged. However, the fact that, in this context, label-free NAD(P)H-FLIM in tissue (and not only in cells and cell cultures – as already cited by the authors) has successfully been employed to answer questions regarding differentiation pathways as well as chronic-inflammatory pathogenesis must be included: Stringari et al, 2012, *Biophys. J.*, Stringari et al, 2013, *Sci.Rep.*, Mossakowski et al, 2015, *Acta Neuropath.*, Radbruch et al, 2016, *Front. Immunol.* Otherwise, the manuscript leads to the erroneous

impression that only by measuring the bulk endogenous NAD(P)H and FAD signal information on the cellular metabolic function can be monitored.

Concluding, I recommend the present manuscript for publication after the authors revise the manuscript commenting on the mentioned issues.

Reviewer #3 (Remarks to the Author):

The article describes the integration of endogenous fluorescence and harmonic microscopy with a single excitation source. A special amplified system with optimized characteristics (low repetition rate, short pulse duration, and 1110 central wavelength) allowed imaging of 4 different modalities with minimal photo damage. Using these modalities, various examples show that tumour biology studies can be performed, leukocytes can be imaged with minimal invasiveness.

All in all, these modalities have been extensively discussed by others for nearly 20 years, and what is new is that they have identified a window where all modalities can be imaged simultaneously. This is mostly a technical achievement.

The images are very impressive, however, it is not shown in this article how well it can operate for real biological studies: we see numerous impressive examples, but it is unclear how this solves problems that are not solved by fluorescent proteins for instance. If the authors intended to show that this could be used in humans, then they should show human data. Imaging leukocyte rolling, adhering and migrating are not new: this has been done with fluorescent markers before. It is not clear how the very dense Figure 4 provides information that is useful to advance tumour biology studies: indeed, they show imaging of several cells in this context, but is this sufficient to extract relevant information for tumour biology? This is not demonstrated since the results remain at the proof of concept stage. The results are an imaged description of what can occur in tissue, but no information about the biological processes are provided.

I would also argue against the comment that SHG and THG are quantitative, since coherent interference will affect the signal distribution and unless the half-sphere (2π steradian) is collected, then it is hard to claim to be quantitative.

I believe this is great technical work, and it shows that this particular source with a compressor and selector may be very useful, but it does not warrant a publication in Nature Communication.

Reviewer #4 (Remarks to the Author):

The authors described an intravital microscopy platform built on a single laser source that is capable of simultaneous imaging with autofluorescence (2FP, 3PF) and multiharmonic (SHG, THG) contrast. They showcased the types of cells that can be analyzed and the dynamic information that can be obtained using the label-free approach. The images presented are quite impressive, although I also have a number of significant concerns.

1. A laser source with very similar characteristics (1150 nm, 11 MHz, 86 fsec) has been described by Huang et al (J Biomed Opt 2017; 22(3), 036008), who also used their system to perform intravital imaging with combined SHG and THG contrast. The novelty of the present work is somewhat diminished by the publication of Huang et al, which should be cited and the relative merits of the two approaches (supercontinuum vs soliton generation) should be discussed.

2. Although SLAM microscopy is "integrated" in the sense that 2PF, 3PF, SHG, and THG can all be acquired at the same time, the manuscript as written lacks true integration in the scientific sense. Beyond showing that different cell types have different optical signatures (different shades of color in the four-channel images), the authors failed to show substantial new insights that are gained from combining information from all these channels. The true power of SLAM microscopy, in my opinion, is the ability to capture cellular dynamics (mostly from THG) and their metabolic profiles (from 2PF and 3PF) at the same time. This opens up a number of interesting questions that can be investigated. For example,

- do activated leukocytes have a different metabolic signature compared to non-activated leukocytes?
 - do leukocytes change their metabolic status when migrating vs. when arrested or forming clusters?
 - do vascular endothelial cells exhibit different metabolic profiles under different flow conditions?
- In my view the manuscript will be strengthened significantly by a more in depth study into any one of these questions.

3. Middle of Page 7, "a ratiometric method using the 2PAF (FAD) and 3PAF (NADH) channels could potentially serve as an indicator of redox states for cells in vivo". This is an important point. However, 2PAF and 3PAF scale differently with laser intensity. How do the authors propose to overcome the fact that the excitation intensity may change from location to location (due to heterogeneity in tissue absorption and scattering), or from superficial to deeper imaging sites, resulting in changing 2PAF/3PAF ratios that does not faithfully report the redox ratio?

4. Videos 4 and 5 shows the formation of several clusters of leukocytes during the 70 min time lapse imaging period. What is triggering the cluster formation? If the clusters are formed in response to the tumor growth, they will most likely have formed during the 4-5 week period after tumor induction and prior to imaging, and not during the 70 min imaging period. Is it possible that the imaging condition itself is causing the cluster formation, indicating some kind of tissue damage?

5. Cells that are aligned along a vascular structure could very well be endothelial cells as the authors suggested. However, they could also be perivascular cells. Additional data, perhaps with the aid of labeling, are needed to rigorously establish the identity of these cells. (There is nothing wrong with using labels to validate results from label-free imaging).

6. The image quality does not appear to be high enough to conclude that cells in the vascular structure are "round biconcave disks with ...no nucleus ..." (Figure 2h). Also, how is it possible to image flowing blood cells without motion blurring? If I am not mistaken, images are acquired at 0.5 f.p.s. so the flow rate would need to be extremely slow in order for the blood cells not to appear blurred.

7. Related to the point above, is a cover glass used for imaging the surgically exposed tumor tissue? If so, what is done to ensure that the pressure from the cover glass does not cause the blood flow to stall (Figure 2f)? Is it possible that the more superficial blood vessel has no flow because it is more sensitive to pressure exerted by the cover glass?

Minor concerns:

8. Please explain the meaning of "resonantly enhance the THG process" (bottom of page 8).

9. It is difficult to tell if the cell in Figure 4h is really undergoing reverse migration because it is not clear if the cell is outside the blood vessel in the first frame.

10. What is D in Figure 4j?

11. Supplementary Fig 1b. Which graphs are for 10 MHz and which are for 80 MHz?

Response to Review Comments

We would like to thank the editor and the four reviewers for finding time to go through our paper and providing very valuable comments and suggestions. We have acquired new data and revised the paper to address all of the comments/suggestions. The manuscript is now improved in clarity and completeness as a result of the feedback from the reviewers.

The important corrections and additions have been highlighted in yellow in the revised manuscript, to facilitate the re-review.

Reviewer #1 (Remarks to the Author):

This is a very interesting study demonstrating a label-free approach to generating in vivo images differentiating multiple cell types in vivo using non-linear methods and a single excitation beam, and a simultaneous multichannel approach. It shows the capacity to differentiate between normal stroma and tumour cells, the vasculature and immune cells all in the absence of exogenous staining or endogenously-expressed reporter proteins. From an immune cell perspective, the videos convincingly show the capability of the system to detect slow rolling and crawling in blood vessels, and impressively even extravascular leukocyte migration in the tumour-associated tissue, in a label-free system.

Response: We thank the reviewer for his/her positive comments and review of our manuscript.

Comment:

A devil's advocate view would be that intravital microscopists for many years used transillumination in tissues such as the mesentery or cremaster muscle to study leukocyte dynamics in and out of vessels without using staining/reporter proteins, and were able to differentiate leukocytes, endothelial cells, mast cells etc. This work provides an extension of this by doing it in 3 dimensions in opaque organs, which is clearly an important advance and should be emphasised.

Response: We'd like to thank the reviewer for this insightful comment. This particular point has added in the modified manuscript (Page 13).

However the authors should comment on the ability of this system to detect different types of leukocytes. The capacity of the SLAM system to detect neutrophil granulocytes is stated to be due to the abundant lipid bodies these cells possess. However, other important immune cells, particularly T cells, carry few lipid granules. How well does this system go in detecting these? Interestingly in Video 3, other leukocytes can be seen, as negative/unstained shapes adhering in the blood vessel, indicating that whatever type of leukocyte this is, it is not being (positively) detected by this system, and presumably would not be visible outside the vasculature. Are these also granulocytes, or might they be other cell types? It would be interesting to examine for example

a lymph node with this system in that this organ is full of lymphocytes and therefore would give an idea of the capacity of this approach to detect other types of immune cells. If it is less effective for these cells, that would be useful information.

Response: We sincerely thank the reviewer for this helpful suggestion and agree that it would be interesting to investigate the ability of SLAM microscopy to detect other types of leukocytes. Therefore, as suggested by the reviewer, we examined a lymph node with SLAM microscopy to test the efficiency of this system for detecting other types of leukocytes, especially lymphocytes.

Response Figure 1a is a full view of an entire rat lymph node, which consists of multiple lymphoid follicles filled with lymphocytes. Reticular fibers (SHG, green), lymph vessels (THG, magenta), and adipocytes (THG/magenta at the periphery and 3PF/cyan in the center) can be seen between or surrounding the lymphoid follicles. The zoomed-in image shows the more refined morphology of individual lymphocytes. As shown in Response Figure 1b, these densely packed lymphocytes are consistently visualized as 6-8 μm hollow-core THG-visible (magenta) spheres, which is remarkably different in both shape and THG intensity from the THG-bright neutrophils (Fig. 2a). The significantly lower THG signals by lymphocytes is consistent with their biological structure since they carry few lipid granules, as pointed out by the reviewer.

Response Figure 1c-e shows leukocytes with varying shapes and THG signals at different depths near a lymphatic vessel. At the depth of 50 μm from the imaging surface (Response Figure 1c), the image revealed densely packed lymphocytes (within white dashed circle) both inside and outside the vessel. Cells within the yellow dashed circle, which are located on top of the lymphoid follicles, exhibited typical morphological features of macrophages, including irregular cell shape and unusually large size ($\sim 20 \mu\text{m}$). As the image plane is placed deeper into the lymphoid follicles, we can see more well-defined lymphocytes on both sides of the vessel (Response Figure 1d-e). In Response Figure 1e, cells within the red dashed circle displayed significantly stronger THG signals, slightly larger size (12-15 μm), and a multi-lobed nucleus, which are the typical features of neutrophils.

For the question regarding the negative shapes adhering to the vessel wall, we agree with the reviewer that they are probably not granulocytes given their significantly weaker THG signals. Based on experiments on lymph node (Response Figure 1), these cells are very likely to be lymphocytes considering their THG intensity and size. They should still be visible outside the vasculature because the optical heterogeneity required for THG contrast remains (as weak as it is) in the extravascular space, as shown in the fresh lymph node image.

In summary, from all these new results, we can conclude that SLAM microscopy is able to detect and identify lymphocytes based on their unique hollow-core sphere-like morphology, small size

(6-8 μm), and relatively weak THG signals. The low THG detection efficiency can be attributed to the fact that lymphocytes carry few lipid granules compared to other leukocytes.

Response Figure 1. Rat lymph node imaging by *ex vivo* SLAM microscopy. (a) Full view of the entire lymph node ($2.6 \times 2.6 \text{ mm}^2$), which consists of multiple lymphoid follicles filled with B lymphocytes. Between the follicles is the cortex where T cells reside. Regions of interest are marked with white dashed squares and magnified in b-f. (b) Lymphocytes are consistently visualized as 6-8 μm hollow-core THG-visible (magenta) spheres, which is remarkably different in both shape and THG intensity from the THG-bright neutrophils (Fig. 2a). (c-e) Images acquired from different depths of a lymphatic vessel, with different depths displaying leukocytes of varying shapes and THG intensities. Cells within white circles are mostly lymphocytes, featured by their small size (6-8 μm), hollow-core sphere-like shape and relatively weak THG signal. Cells within yellow circles are identified as macrophages, considering their larger size ($\sim 20 \mu\text{m}$) and irregular shape. Cells within the red circle are likely neutrophils, given their size (12-15 μm) and multi-lobed nuclei.

Also, would it be possible to confirm some of the observations in regards to immune cell identity by performing similar experiments using established approaches making use of labelling to

confirm the identity of the immune cells examined (this may not be straightforward in a rat-based system)?

Response: We fully understand the reviewer's concern and conducted a series of experiments to confirm the identity of the immune cells shown in the figures.

First, we injected a CD45-based marker (Monoclonal antibody OX1, eBioscience) into a living rat and acquired images of the tumor microenvironment before and after labelling (Response Figure 2). Before labelling, the blue channel visualized mostly NADH-rich vesicles while the yellow channel showed FAD-rich stroma cells and the purple channel picked up signals from THG-visible hollow spheres (6-8 μm in size), which are supposedly lymphocytes. After labelling, stained leukocytes should show up in the blue channel while the assignment for the other channels remain the same. As shown in Response Figure 2, there are three groups of cells in this dataset: 1) yellow stroma cells that remain yellow after staining, which validates the specificity of both THG and CD45 for leukocytes; 2) leukocytes with overlapping THG and CD45 signals, which confirms the identify of these THG-visible cells; 3) leukocytes with CD45 but no THG signals, which suggests that SLAM microscopy only visualizes subpopulations of leukocytes. This discrepancy might be explained by the new data on the rat lymph node (Response Figure 1), which shows that THG is not highly efficient for detecting all lymphocytes as they carry few lipid granules. It could also be caused by technical issues, such as a slight change of the imaging depth before and after staining due to the injection procedure.

As pointed out by the reviewer, anti-rat antibodies specific to different immune cells are challenging to obtain and apply, we resorted to validating with lymph node imaging, as shown in Response Figure 1. Combining these images with the staining results, we are confident to say that we can identify leukocytes based on their unique THG optical signature and potentially differentiate neutrophils, macrophages and leukocytes based on their differences in size, morphology and THG signal intensity.

In summary, these results demonstrate that SLAM microscopy has high specificity and sensitivity for leukocyte detection. A more in-depth characterization will be performed in a more focused study in the future.

Response Figure 2. Tumor microenvironment of a living rat before and after CD45 staining by SLAM microscopy. The blue channel was assigned to NADH before staining and assigned to CD45 after staining. The image after staining shows cells with overlapping THG (magenta) and CD45 (blue) signals, demonstrating the specificity of THG signals. Scale bar: 50 μ m.

Minor comments:

Intravital microscopy has been in use for over 100 years, and gained prominence in the 1980s, so the opening sentence should be modified.

Response: We thank the reviewer for noticing this. The opening statement has been corrected in the revised manuscript (Page 3).

The statement on p8 about the limited capabilities of current methods to provide morphological/molecular information on different types of immune cells is not a true representation of the current state of the field – it is possible now to definitively identify different immune cell subsets, even rare ones like regulatory T cells, as well as studying their surface phenotype (e.g. adhesion molecule expression) and metabolic state (e.g. ROS production). This sentence should be rewritten to reflect this.

Response: We thank the reviewer for pointing out this misleading statement. This sentence has been rewritten to better reflect the current state of the field (Page 8).

There are some technical details I think it would be beneficial to add:

- Depth of imaging. It is stated that images were taken from 20-200 μm below the surface. It would be useful for assessing the utility of this approach if example images could be provided at different defined depths, along the lines of the first video (out to 115 μm). Providing a visual demonstration of where these signals start to drop out, and how each of the types of signal is individually affected would be useful.

Response: We thank the reviewer for this helpful suggestion. Representative images along the lines of the first video have been added as Supplementary Figure 4 and referenced in the main text on Page 7. Based on the raw contrast, the signals start to significantly drop out around 100 μm . Nevertheless, after adjusting contrast and brightness in imaging software (FIJI, National Institutes of Health), most stromal and cellular structures are still visible at the depth of 100 μm .

Supplementary Figure 4. Representative images at different depths within the tumor microenvironment (Fig2. f-h and Supplementary Movie 1). (a) Images with consistent data range and color scale throughout the increasing depth of the image plane. (b) In order to better visualize structures, contrast and brightness were optimized (FIJI, National Institutes of Health) for images acquired at deeper locations.

- Time resolution – how quickly can these 3D images be acquired?

Response: We added a more detailed description of the temporal resolution in the manuscript (Page 15-16). Each image took 2 seconds to acquire. Therefore, the 8 depth-resolved images shown in Supplementary Fig. 4 took approximately 17 seconds. For the raw video, which was sampled with 75 slices along a depth range of 75 μm , acquisition required approximately 160 seconds.

- What are the bandwidth characteristics of the filters splitting the emitted signal into the respective PMTs?

Response: All the bandwidth characteristics of filters were listed in the table inserted in Fig.1. To make this more visible to readers, we added a short description in the revised manuscript (Page 15).

Reviewer #2 (Remarks to the Author):

The manuscript of You et al presents simultaneous label-free auto-fluorescence microscopy (SLAM) and its use for intravital imaging in rodents. The innovative approach adapts and applies the excitation source, the authors developed and published in the context of histopathology, to the demanding field of intravital microscopy. The power of the technology is demonstrated in tumor environment context – a field of highest scientific impact – showing its advantage regarding the acquisition speed as well as the possibility to monitor tissue and cellular metabolic function in vivo. I expect that the here presented technology will open unprecedented perspectives for the label-free intravital functional imaging for various branches of biomedical research, not being limited to the here discussed application.

Response: We thank the reviewer for her/his positive comments and review of our manuscript. We certainly believe that this new technology will impact many areas of biomedical research.

While the entire experimental work has been carefully performed and the results are well interpreted, in order to make the manuscript thoroughly valid, the SLAM technology needs to be better placed in the context of already existing intravital imaging technologies (both label-free or labeling-based):

The authors discuss the possibility to perform single wavelength excitation at a time in a multi-photon microscope and correctly refer to the work from the Debarbieux lab regarding sequential single-wavelength excitation multiplexed in vivo imaging. In this context, the four-color label-free intravital imaging after excitation at 1110 nm (10 MHz) is the unique way to perform simultaneous imaging of NAD(P)H, FAD next to SHG and THG. However, in the manuscript, the fact that simultaneous multiple wavelengths excitation in a multi-photon microscope has been successfully performed and is now an established intravital imaging technique is completely omitted (Herz et al, Biophys. J., 2010; Mahou et al, Nat. Methods, 2012; Rakhymzhan et al, Sci. Rep., 2017). For a comprehensive understanding, the authors should comment in more detail on the possibility of using a combined simultaneous excitation at 740 nm and 1110 nm (nowadays available in standard multi-photon microscopes, from Ti:Sa lasers combined with optical parametric oscillators or amplifiers) instead of 1110 nm alone using their custom-made excitation source. An argument for their excitation source is surely the lower repetition rate of 10 MHz together with the relatively narrow pulse width– which allow for a more effective excitation, better photon management and, thus, higher acquisition speeds and, eventually, lower photobleaching and photodamage. Still, this advantage has not been demonstrated. At least a comment on this issue and on the possibility to use well-established, commercially available laser amplification (pulse picker / Pockels cells etc.) is necessary, beyond the already cited solutions applied in exclusively label-free IVM. Hence, the following questions arise: (1) Why is a custom-made, not easily available single-wavelength excitation better than multiple-wavelength excitation, available in standard multi-

photon microscopes? (2) Why is the custom-made excitation source better than the “old” solutions for the “old” requirement of reducing the laser repetition rate and increasing pulse energy and peak power (i.e. narrow pulse width) to achieve a higher signal at low mean laser powers.

Response: We thank the reviewer for these insightful suggestions. These points are important to illustrate the importance of this work and have now been incorporated more completely into the manuscript (Page 14). A more detailed discussion is provided as followed.

We fully agree with the reviewer that we should emphasize the advantages of this fiber source in comparison to the existing technologies. If we were to compare our fiber source with the excitation at 1110 nm provided by a Ti:Sapphire laser combined with OPO, the latter has much lower peak intensity (80 MHz vs. 10 MHz, 140-400 fs vs. 35 fs) and will lower the signal generation efficiency by three orders-of-magnitude due to the inverse second-order dependence on pulse duration and laser repetition rate in third-order nonlinear processes (Supplementary Fig. 1). Therefore, we believe the fiber source with a widely coherent spectrum together with a lower repetition rate is critical for achieving non-perturbative and near-real-time recording of biological events. In addition, the high peak power of this single-source single-excitation configuration allows balanced signal detection between second- and third-order nonlinear processes, so that comparable two-photon and three-photon signals can be acquired.

The reviewer is also correct that simultaneous multiple wavelength excitation in a multi-photon microscope has been successfully performed and widely applied in both label-free and labeling-based work. However, the major challenges presented in these two fields of microscopy are vastly different from each other. In labeling-based work, the main challenge of simultaneous multicolor imaging is to come up with a set of efficiently-binding, non-interfering and complementary markers. In a label-free system, however, the main challenge for achieving simultaneous contrasts originates from the weak endogenous contrast, and thus the demanding excitation/emission requirements. As a result, implementation of SLAM microscopy heavily relies on optimizing excitation and emission conditions to maximize the SNR of each channel while avoiding crosstalk. For this reason, the aforementioned combined excitation (740 nm and 1110 nm) would be perfect for illuminating a carefully-designed fluorescent marker set, as shown in previous literature [Ref: Expanding two-photon intravital microscopy to the infrared by means of optical parametric oscillator, 2010, J. Herz et al; Multicolor two-photon light-sheet microscopy, 2014, P. Mahou et al; Synergistic strategy for multicolor two-photon microscopy-application to the analysis of germinal center reactions *in vivo*, 2017, A. Rakhymzhan et al]. However, this combined excitation would not work for simultaneous detection of autofluorescence and multiharmonic generation as 740 nm will generate strong SHG signals at 370 nm, which would completely overlap with or even overwhelm the THG channel (370 nm) at 1110 nm excitation. Therefore, a combined excitation would not benefit this label-free imaging scheme.

The authors point out at the possibility to use their technology to quantify cellular metabolism and to differentiate between oxidative phosphorylation and glycolysis – a reliable marker to distinguish between highly active tumor cells and normal/healthy cells in tissue. The possibility to monitor tissue and cell function is invaluable and is greatly acknowledged. However, the fact that, in this context, label-free NAD(P)H-FLIM in tissue (and not only in cells and cell cultures – as already cited by the authors) has successfully been employed to answer questions regarding differentiation pathways as well as chronic-inflammatory pathogenesis must be included: Stringari et al, 2012, Biophys. J., Stringari et al, 2013, Sci.Rep., Mossakowski et al, 2015, Acta Neuropath., Radbruch et al, 2016, Front. Immunol. Otherwise, the manuscript leads to the erroneous impression that only by measuring the bulk endogenous NAD(P)H and FAD signal information on the cellular metabolic function can be monitored.

Response: We thank the reviewer for noticing this. In the revised manuscript, references of important work on FLIM have been added as suggested by the reviewer (Page 4).

Concluding, I recommend the present manuscript for publication after the authors revise the manuscript commenting on the mentioned issues.

Response: We thank the reviewer for the thorough evaluation and valid suggestions. We believe the manuscript is much improved after taking these suggestions into consideration in the revised manuscript.

Reviewer #3 (Remarks to the Author):

The article describes the integration of endogenous fluorescence and harmonic microscopy with a single excitation source. A special amplified system with optimized characteristics (low repetition rate, short pulse duration, and 1110 central wavelength) allowed imaging of 4 different modalities with minimal photo damage. Using these modalities, various examples show that tumour biology studies can be performed, leukocytes can be imaged with minimal invasiveness.

All in all, these modalities have been extensively discussed by others for nearly 20 years, and what is new is that they have identified a window where all modalities can be imaged simultaneously. This is mostly a technical achievement.

The images are very impressive, however, it is not shown in this article how well it can operate for real biological studies: we see numerous impressive examples, but it is unclear how this solves problems that are not solved by fluorescent proteins for instance. If the authors intended to show that this could be used in humans, then they should show human data. Imaging leukocyte rolling, adhering and migrating are not new: this has been done with fluorescent markers before. It is not clear how the very dense Figure 4 provides information that is useful to advance tumour biology studies: indeed, they show imaging of several cells in this context, but is this sufficient to extract relevant information for tumour biology? This is not demonstrated since the results remain at the proof of concept stage. The results are an imaged description of what can occur in tissue, but no information about the biological processes are provided.

Response: We thank the reviewer for the review of our manuscript and the important concerns that are raised.

For the first criticism, we believe this paper provides promising evidence that this label-free technology can solve problems that are not solved by fluorescent proteins. First of all, as pointed out by other reviewers, one of the most attractive attributes of SLAM microscopy is the capability to capture cellular dynamics/interactions (mostly from THG and SHG) and their metabolic profiles (from 2PF and 3PF) in one shot, which opens up a lot more possibilities in biomedical research. To further demonstrate this capability (as requested by other reviewers) with a more specific example, we added an experiment to investigate the changes in the metabolic status of cells when they are migrating and when they are forming a cluster (Page 11 and new Supplementary Figure 6). This type of investigation is straightforward for SLAM microscopy but can be very complicated for label-based technologies [Ref: Imaging ROS signaling in cells and animals, 2013, X. Wang et al]. Secondly, for the comparison between label-based and label-free work, we believe the advantages of a label-free imaging technology were overlooked by the reviewer. Methods based on fluorescence proteins are fundamentally limited by the complicated and sometimes unpredictable tissue distribution of the exogenous markers, unexpected or unknown disturbance of biological or physiological functions, and unavoidable artifacts of non-specific labeling. If we

have a way to generate comparable imaging quality (if not better) without these limitations, this capability will be seen as progress itself and will serve as an attractive complementary approach to existing intravital microscopy methods.

Also, we respectfully do not agree that the significance of this work is diminished because we just “show imaging of several cells in context” or “remain at the proof of concept stage”. First, we were able to provide unique insights into the metabolic shifts in cells with different dynamics or surroundings, as mentioned in the previous paragraph. Second, visualizing cells in context has been an essential endeavor and challenge in the life sciences [Ref: Intravital microscopy: new insights into metastasis of tumors, 2011, E. Beerling et al; Dissecting tumour pathophysiology using intravital microscopy, 2002, R. K. Jain]. Efforts to improve this should not be considered trivial. Third, showing the possibility to monitor tissue and cellular metabolic function and interactions *in vivo*, including tumor cells, vascular endothelial cells, blood cells, immune cells, and their surrounding ECM, is very relevant and useful for advancing tumor biology, as shown by a vast range of previous studies [Ref: Tumor microvasculature and microenvironment: novel insights through intravital imaging in pre-clinical models, 2010, D. Fukumura et al; Three-dimensional microscopy of tumor microenvironment *in vivo* using optical frequency domain imaging, 2009, B. J. Vakoc]. Finally, we would like to emphasize that our intention is to demonstrate an enabling technology and show its potential impact in various branches of biomedical research, especially in the context of the living tumor microenvironment. A comprehensive large-scale hypothesis-driven biological study is beyond the scope of this manuscript, but will be conducted in future studies. Future studies will include systematic quantitative analysis of these SLAM microscopy images/videos from longitudinal rat mammary tissue/tumor changes via an imaging window, as well as changes in breast organoids from human subjects following administration of various chemotherapeutic drugs.

The reviewer is correct that human data is important for demonstrating clinical potential. However, we did not include or talk about human data in this paper as the focus of this paper is intravital imaging. We felt that the most significant breakthrough is the ability to capture various cellular dynamics with molecular and metabolic profiling capacity in unperturbed complex living systems.

[REDACTED]

In summary, we agree with the reviewer that new discoveries in tumor biology are fundamental pursuits for any scientist in this field. However, better tools, including imaging tools, are the key to enable more discoveries in the future. The novelty and the potential impact of this paper originates from the capability to simultaneously detect metabolic activity (autofluorescence from

FAD and NADH) and molecular structures (SHG/THG), which enables straightforward functional and structural imaging of various cellular events in the authentic unperturbed tumor microenvironment *in vivo*, as demonstrated in the Results section. We believe the unique system and the promising results presented in this manuscript will have immediate and lasting impact in

[REDACTED]

[REDACTED]

[REDACTED]

[REDACTED]

[REDACTED]

[REDACTED]

I would also argue against the comment that SHG and THG are quantitative, since coherent interference will affect the signal distribution and unless the half-sphere (2π steradian) is collected, then it is hard to claim to be quantitative.

Response: The reviewer is correct that coherent interference significantly complicates the extraction of molecular information in SHG and THG processes. As a result, whether these two processes can be used as quantitative measurements has remained a controversial topic. Therefore,

we have modified our discussion to make readers aware of this controversy (Page 12). A more-detailed review is provided as follows.

For SHG processes, the interference pattern of the transmitted SHG signals can be described as $\sin^2(2\pi\Delta nL/\lambda_{ex})$, where Δn is the dispersion of the sample and L is the sample scattering length. Therefore, SHG intensity spectral dependence has been proposed to accurately extract quantitative tissue information (e.g. dispersion and scattering length) [Ref: Imaging cells and extracellular matrix *in vivo* by using second-harmonic generation and two-photon excited fluorescence, 2002, A. Zoumi et al]. For backward detection, SHG in tissue is usually described as quasi-coherent considering the imperfect phase matching condition. Using a more general model of SHG by considering a relaxed phase-matching condition, the initial SHG intensity is determined by the second order nonlinear susceptibility χ^2 [Ref: Second harmonic generation microscopy for quantitative analysis of collagen fibrillar structure, 2012, X. Chen et al]. Although the absolute χ^2 coefficients are challenging to extract, the relative conversion efficiencies can be derived based on SHG intensity measurements [Ref: Quantitative second harmonic generation of the diseased state osteogenesis imperfecta: experiment and simulation, 2008, R. LaComb et al]. R. LaComb et al showed that that the bone, skin, and tendon from osteogenesis imperfecta mouse were substantially distinct from the wild-type by using differences in the intensity as well as morphological differences from SHG images. Therefore, although it is very complex to decouple all the factors that contribute to SHG creation and propagation, it is possible to extract the bulk optical parameters and provide quantitative and statistical distinction in biological samples.

For THG processes, quantification is even more challenging due to its higher-order nonlinearity and sensitivity to slight changes in imaging conditions. Under phase-matching conditions, the signal is proportional to $\kappa|\alpha_1 - \alpha_2|^2 \langle I_{ex}^3 \rangle$, where κ represents sample geometry, and $|\alpha_1 - \alpha_2|$ reflects the optical properties of the two media [Ref: Quantitative characterization of biological liquids for third-harmonic generation microscopy, 2007, D. De'barre et al]. Under a tight focus ($NA > 0.8$, $NA = 1.15$ in this paper), α approaches χ^3 [Ref: Imaging lipid bodies in cells and tissues using third-harmonic generation microscopy, 2006, D. De'barre et al]. With the known nonlinear susceptibility of biological solutions, it is possible to extract the optical properties of biological structures based on THG intensities. D. De'barre et al have directly used this endogenous signal to quantify the size distribution of lipid bodies under different physiological conditions in isolated hepatocytes as well as in unstained fresh tissue.

I believe this is great technical work, and it shows that this particular source with a compressor and selector may be very useful, but it does not warrant a publication in Nature Communication.

Response: We thank the reviewer for his/her appreciation of the technology. However, as stated before, we believe the system and the unique results presented in this manuscript have the requisite

originality and scientific impact that is worth sharing with the audience of Nature Communications. We anticipate SLAM microscopy will be an attractive alternative or complementary approach to existing intravital microscopy in various applications owing to its label-free nature, relative simplicity, versatility, and rich molecular profiling capability. We firmly believe that this manuscript is appropriate for publication in Nature Communications.

Reviewer #4 (Remarks to the Author):

The authors described an intravital microscopy platform built on a single laser source that is capable of simultaneous imaging with autofluorescence (2FP, 3PF) and multiharmonic (SHG, THG) contrast. They showcased the types of cells that can be analyzed and the dynamic information that can be obtained using the label-free approach. The images presented are quite impressive, although I also have a number of significant concerns.

Response: We thank the reviewer for his/her thorough review of our manuscript and the important concerns that are raised. We believe our new experiments and added discussion should help address these concerns.

1. A laser source with very similar characteristics (1150 nm, 11 MHz, 86 fsec) has been described by Huang et al (J Biomed Opt 2017; 22(3), 036008), who also used their system to perform intravital imaging with combined SHG and THG contrast. The novelty of the present work is somewhat diminished by the publication of Huang et al, which should be cited and the relative merits of the two approaches (supercontinuum vs soliton generation) should be discussed.

Response: We thank the reviewer for raising this point. We agree it is important to cite this work (added on Page 4) as their source has similar parameters in terms of the fiber model, excitation wavelength, and repetition rate, and that they have demonstrated combined SHG and THG imaging. However, we do not think the novelty of the present work is diminished by this cited work. Despite the novelty of their source, combined SHG and THG imaging is relatively common in long-wavelength (LW) multiphoton imaging platforms, with a detailed review in the Introduction on Page 4. As discussed in the Introduction (Paragraph 3, Page 4), the main challenge for integrating these four contrasts in a LW platform is the significantly lower absorption cross sections of the intrinsic fluorophores at a longer excitation wavelength, which was not tackled or addressed in the work by Huang et al. Although they also used a lower rep rate (11 MHz), their source would hardly be sufficient for real-time simultaneous imaging of FAD, NADH, SHG and THG, considering that their longer wavelength (1150 nm vs. 1110 nm) excitation efficiency for FAD/NADH drops exponentially at longer wavelengths, and for their relatively longer pulse duration (86 fs vs. 35 fs).

We fully understand the reviewer's concern that the merits of supercontinuum (SC) and self-frequency shifted soliton generation (SSFS) should be compared, and we have added this discussion in the revised manuscript (Supplementary Note 2). A general discussion of the difference is available in multiple review papers [Ref: Supercontinuum generation in photonic crystal fiber, 2006, J. M. Dudley et al; Coherent fiber supercontinuum for biophotonics, H. Tu et al; Soliton self-frequency shift: experimental demonstrations and applications, 2008, J. H. Lee et al] and we will focus on the differences that directly affect the application to SLAM microscopy

in this paper. The most obvious difference is that SC has a broader spectrum than SSFS (300 nm vs. 20-60 nm). The 60-nm bandwidth employed by SLAM microscopy is necessary to obtain an ultrashort pulse to support efficient excitation of FAD and NADH at 1110 nm. The other difference is the complexity in optimizing the excitation window for multiphoton imaging. In SC generation, after a wide and stable SC is produced, choosing an excitation window is straightforward and easy as we only need to select the spectral region of interest out of the available wavelengths across the spectral bandwidth. In SSFS, however, choosing a different wavelength requires re-adjustment of pump power and fiber length as well as customization of the SHG crystal (BIBO and MgO:PPLN).

In summary, we acknowledge the contribution of the cited work but do not think the novelty of this work is diminished in this way. The novelty and the potential impact of our paper originates from the capability to simultaneously detect auto-fluorescence (FAD and NADH) and SHG/THG, which enables straightforward functional and structural imaging of various cellular events in the authentic tumor microenvironment *in vivo*. The demonstrated fiber source with a widely coherent and uniform supercontinuum at lower repetition rate is unique and essential for making this possible. Finally, compared to settings in the cited work, this configuration is simple and flexible, which makes it attractive for various applications, especially for clinical translation in the future.

2. Although SLAM microscopy is "integrated" in the sense that 2PF, 3PF, SHG, and THG can all be acquired at the same time, the manuscript as written lacks true integration in the scientific sense. Beyond showing that different cell types have different optical signatures (different shades of color in the four-channel images), the authors failed to show substantial new insights that are gained from combining information from all these channels. The true power of SLAM microscopy, in my opinion, is the ability to capture cellular dynamics (mostly from THG) and their metabolic profiles (from 2PF and 3PF) at the same time. This opens up a number of interesting questions that can be investigated. For example,

- do activated leukocytes have a different metabolic signature compared to non-activated leukocytes?
 - do leukocytes change their metabolic status when migrating vs. when arrested or forming clusters?
 - do vascular endothelial cells exhibit different metabolic profiles under different flow conditions?
- In my view the manuscript will be strengthened significantly by a more in depth study into any one of these questions.

Response: We thank the reviewer for these insightful suggestions. We agree with the reviewer's comment that one the most powerful attributes of SLAM microscopy is the capability to capture cellular dynamics and their metabolic profiles at the same time. We also agree that the manuscript will be strengthened by a more in-depth study into any of the above questions. Therefore, we have investigated the first two proposed research questions and added these insights into the revised manuscript (Page 11 and Supplementary Figure 6 as shown below).

Q2: Do leukocytes change their metabolic status when migrating vs. when arrested or forming clusters?

This is a great question as ongoing inflammation and a triggered immune response is expected to be accompanied by significant changes in metabolic activity [Ref: Metabolic shifts in immunity and inflammation, 2010, D. J. Kominsky]. We analyzed data from Supplementary Movie 5 to see whether leukocytes change their metabolic status when migrating vs. when forming clusters. In the representative images shown below, the cells with blue marks are those that are swarming and forming the clusters while the cells with red marks are those that are migrating toward a cluster. Compared to the migrating cells, the clustering cells appear slightly more blueish, which indicates higher NADH and a lower redox ratio. Besides this visual observation, quantification of the redox ratio, $FAD/(FAD+NADH)$, was also performed on two groups of manually segmented cells (blue for the cytoplasm of clustering cells and red for the cytoplasm of migrating cells). The redox ratio significantly ($p < 0.0001$ by Student's t-test) decreased in the clustered cells compared to the migrating cells, indicating an increase in metabolic activity as well as a hypoxic environment [Ref: *In vivo* multiphoton microscopy of NADH and FAD redox states, fluorescence lifetimes, and cellular morphology in precancerous epithelia, 2007, M. C. Skala et al]. This dramatic decrease of the redox ratio is likely due to the combined effect of the tremendous amount of energy consumed in cell transit and the diminished delivery and/or availability of oxygen at the center of the cluster. [Ref: Metabolic shifts in immunity and inflammation, 2010, D. J. Kominsky].

Supplementary Figure 6. Metabolic profiling of cells with different motion dynamics. (a) Image (2PAF (yellow) and 3PAF (blue)) was extracted from the end of Supplementary Movie 5 and shows clustered leukocytes and migrating leukocytes at the same site. (b) A mask marks two sets of representative cells – clustered leukocytes (blue) and migrating leukocytes (red). (c) Scatterplot of the redox ratio for each cell in the clustered group and the migrating group. The redox ratio significantly ($p < 0.0001$ by Student's t-test) decreased in the clustered cells compared to the migrating cells. Scale bar: 50 μm .

As a side note, we tried to perform the same analysis on the cells that were arrested in the vessel. However, an accurate redox ratio analysis is compromised by the strong and broad fluorescence emitted from hemoglobin in the perfusing blood vessel. Nevertheless, the comparison between clustering cells and migrating cells sufficiently demonstrates the potential of SLAM microscopy to provide metabolic profiling for individual cells with dynamic details in the extravascular space.

In short, we believe these added experiments and discussions 1) further demonstrate the power of SLAM microscopy to capture cellular dynamics and metabolic changes at the same time; 2) answers the reviewer's question about the differences in the metabolic status between clustered cells and migrating cells, as well as the differences between activated and non-activated leukocytes; and therefore, 3) makes this manuscript more comprehensive and complete.

3. Middle of Page 7, "a ratiometric method using the 2PAF (FAD) and 3PAF (NADH) channels could potentially serve as an indicator of redox states for cells in vivo". This is an important point. However, 2PAF and 3PAF scale differently with laser intensity. How do the authors propose to overcome the fact that the excitation intensity may change from location to location (due to heterogeneity in tissue absorption and scattering), or from superficial to deeper imaging sites, resulting in changing 2PAF/3PAF ratios that does not faithfully report the redox ratio?

Response: We thank the reviewer for this insightful comment. Compared to lifetime measurements, a redox ratio measurement is indeed more sensitive to tissue absorption and scattering. We propose to limit this susceptibility by diligent calibration of excitation intensity and imaging depth, which is commonly practiced in multiphoton redox measurements [Ref: Multiphoton redox ratio imaging for metabolic monitoring *in vivo*, 2009, M. C. Skala et al], and customized ratio metrics. Since 2PAF and 3PAF are simultaneously collected and expected to have identical incident power, we could potentially derive the relative concentration of NADH and FAD based on the known power dependence and excitation efficiency of the system (daily calibration data), which should provide a reliable qualitative assessment of the redox state.

4. Videos 4 and 5 shows the formation of several clusters of leukocytes during the 70 min time lapse imaging period. What is triggering the cluster formation? If the clusters are formed in

response to the tumor growth, they will most likely have formed during the 4-5 week period after tumor induction and prior to imaging, and not during the 70 min imaging period. Is it possible that the imaging condition itself is causing the cluster formation, indicating some kind of tissue damage?

Response: We thank the reviewer for this intriguing question. Leukocyte clustering is known to be triggered by various conditions of inflammation, including tumor progression and laser damage. However, we do not think this is due to laser damage as the same experiments have been performed on control animals and no similar phenomena has been observed. In addition, no apparent laser damage is observed at the end of the video (after 70 minutes), which is expected for near IR excitation with an average power of 14 mW. Nevertheless, more controlled experiments are needed to pinpoint the exact trigger for this leukocyte cluster formation in future studies.

5. Cells that are aligned along a vascular structure could very well be endothelial cells as the authors suggested. However, they could also be perivascular cells. Additional data, perhaps with the aid of labeling, are needed to rigorously establish the identity of these cells. (There is nothing wrong with using labels to validate results from label-free imaging).

Supplementary Figure 3. Images of tumor cells, endothelial cells, and red blood cells by SLAM microscopy (a-c) and corresponding H&E-stained histology (d-f). Red arrows in (b,e) point to corresponding cellular features, and dotted red lines in (c) outline the walls of a blood vessel. Scale bar: 50 μm .

Response: We thank the reviewer for this helpful suggestion. Comparisons between SLAM microscopy images and H&E stained tissue sections have been added to the revised manuscript as Supplementary Figure 3. The high degree of similarity in size, morphology, and spatial distribution provides convincing evidence for the identity of tumor cells, vascular endothelial cells, and red blood cells in the images.

6. The image quality does not appear to be high enough to conclude that cells in the vascular structure are "round biconcave disks with ...no nucleus ..." (Figure 2h). Also, how is it possible to image flowing blood cells without motion blurring? If I am not mistaken, images are acquired at 0.5 f.p.s. so the flow rate would need to be extremely slow in order for the blood cells not to appear blurred.

Response: We thank the reviewer for noticing this. We agree "biconcave" can be speculative and we deleted this part from the description of the red blood cells. The part about "round and no nucleus" though should be obvious from the image, especially compared to other cells, which are mostly spindle-shape with a dark center (non-fluorescent nuclei), as shown in Fig. 2c and d.

The reviewer is also correct that the blood cells will be blurry and shaky if they are flowing in the vessel, as shown in Fig. 3 and Fig. 4. We think the clear view (slow flow) of individual blood cells (Fig. 2h) might be attributed to accidental vessel damage during the surgical procedure, which is hard to avoid in skin flap models. We are planning to adopt a longitudinal mammary imaging window model to avoid such artifacts in the future.

7. Related to the point above, is a cover glass used for imaging the surgically exposed tumor tissue? If so, what is done to ensure that the pressure from the cover glass does not cause the blood flow to stall (Figure 2f)? Is it possible that the more superficial blood vessel has no flow because it is more sensitive to pressure exerted by the cover glass?

Response: The coverslip is actually below the tissue as we use a custom-designed inverted microscope. Therefore, the tissue should not suffer from any other pressure other than the effects of gravity on the tissue itself.

Minor concerns:

8. Please explain the meaning of "resonantly enhance the THG process" (bottom of Page 8).

Response: We thank the reviewer for pointing this out. We modified "resonantly" to "coherently" to be more precise and have added references for detailed explanations of the mechanism (Page 9). For a brief explanation, first, neutrophils contain a high concentration of lipids in an aqueous environment, which guarantees the presence of a high $\chi^{(3)}$ (the third-order nonlinear susceptibility) [Ref: Imaging lipid bodies in cells and tissues using third-harmonic generation microscopy, 2006, D. De´barre et al]. Then, because the final THG signal intensity is a coherent superposition of the fields radiated within the excitation volume, the THG signals will be coherently enhanced when the size and organization of the lipid bodies in neutrophils match well with the volume of the beam focus [Ref: Third harmonic generation microscopy of cells and tissue organization, 2016, B. Weigelin; Imaging granularity of leukocytes with third harmonic generation microscopy, 2012, C. K. Tsai].

9. It is difficult to tell if the cell in Figure 4h is really undergoing reverse migration because it is not clear if the cell is outside the blood vessel in the first frame.

Response: We understand the reviewer's concern that it is difficult to tell if the cell is outside the blood vessel in the first frame. The description of this motion as reverse migration is mostly derived from the unusual migration patterns of these leukocytes. Based on the later morphological changes (round to elongated) and migration pattern changes (increasing speed until disappearing with fast perfusing red blood cells), we can see that this leukocyte is undergoing the opposite process of being arrested, which is very likely to be reserve migration. To avoid misleading the

readers, we tuned down our interpretation of reverse migration and rewrote the description of the movement in the manuscript (Page 11).

10. What is D in Figure 4j?

Response: As indicated in the caption, D is the abbreviation for directionality. To make this clearer to readers, we slightly modified the explanation in the caption for Figure 4 (Page 24).

11. Supplementary Fig 1b. Which graphs are for 10 MHz and which are for 80 MHz?

Response: We thank the reviewer for noticing this. Notations have been added to the captions in the revised manuscript.

Other changes:

1. To make it more intuitive, we added a time-lapse counter to Movie 5.
2. Due to the limit of the total characters in subheadings, we changed the first subheading in Result section from “Simultaneous visualization of various cancer-associated stromal cells and vesicles by SLAM microscopy” to “*In vivo* tumor microenvironment by SLAM microscopy”.

REVIEWERS' COMMENTS:

Reviewer #1 (Remarks to the Author):

While the authors have had a reasonable go at addressing the comments I raised, it seems like an opportunity lost that the work they have done to address them has not been incorporated into the manuscript, where it could provide evidence supporting some of their claims.

In the Response Figure 1, the authors show some images of a lymph node showing likely lymphocytes and macrophages (the claimed neutrophils are less clear), illustrating to some extent the capacity of this system to differentiate between cell types.

In the Response Figure 2, the authors show some images showing combined label-free and anti-CD45-labelled images from a tumour. Unfortunately the two frames do not show the exact same region, so they don't allow an optimal comparison to be made. A more simple way forward would be to display the right hand panel twice, once with the anti-CD45 signal turned off, so that the label-free labelling of the anti-CD45-stained cells can be assessed more readily. However, if displayed appropriately, this would make an excellent addition to the manuscript.

It is a pity that these new pieces of data were not included in the manuscript at this revision, as particularly for Response Figure 2, it makes an important point. Indeed it is notable that in the revised supplementary Figures, Supplementary Figure 3 purports to show the same regions of tissue using label-free SLAM microscopy and histology. Using histology it is extremely difficult to be sure that you are visualising the same area of tissue you were examining under an intravital microscope. In contrast the approach attempted in Response Figure 2 allows visualisation of the identical field with and without label-free microscopy (simply by turning on and off the anti-CD45 channel), and is therefore much better than a histological approach, but it is not included in the manuscript! I suggest the authors work to find a way to include and describe particularly Response Figure 2 in the Supplementary data of revised version.

Reviewer #2 (Remarks to the Author):

In the revised version of the manuscript, the authors addressed all my concerns and significantly improved both the introduction and the discussion. I strongly recommend the manuscript for publication.

Reviewer #4 (Remarks to the Author):

I thank the authors for carefully addressing many of the concerns raised in the previous review. There are however several remaining concerns:

1. I agree with Reviewer 3 that imaging leukocyte rolling, adhering and migrating are not new, and suggest placing less emphasis on this part of the paper. Specifically, the two sections headlined "Intravascular leukocyte recruitment" and "Interstitial leukocyte migration" can be combined and shortened significantly. Figures 3 and 4 can be moved to the supplement. Instead, more emphasis should be placed on the new results, namely the simultaneous capturing of cellular dynamics and metabolic profiles, to showcase the novel feature of SLAM microscopy. An expanded version of Supplementary Figure 6, with

a more in depth analysis of the metabolic differences between migrating vs. cluster-forming leukocytes, should be moved to the main figure, accompanied by a new movie showing the tracks of the blue and the red cells so that readers have a better sense of how these cells moved.

2. In response to my previous question about ratiometric analysis, the authors propose to carry out "diligent calibration" and use "customized ratio metrics" (p.20 of the response letter). Please described exactly how this is going be done in the manuscript (not just in the letter) so that the analysis can be reproduced by other labs.

3. Regarding the question what triggered cluster formation, the author presented a reasonable argument that it is probably not due to laser damage (p.21 of the response letter). However, in response to a following concern about the sluggish blood flow, the authors pointed out that it is hard to avoid accidental vessel damage during the surgical procedure. Might the injury be the trigger for the leukocyte clustering? Again please address the issue of surgical damage in the manuscript and not just in the letter so the readers are aware of the potential confounding factors when attempting to reproduce the results.

4. Related to the point above, please restrict the use of the term "non-perturbative", which is used repeatedly in the manuscript.

5. Regarding the cell undergoing reverse transmigration (p.23 of the letter and p.11 of the manuscript), since it is difficult to tell if the cell was outside the blood vessel in the first frame, it is just as likely that the cell was initially arrested inside the blood vessel and subsequently detached and disappeared in the flow, without undergoing transmigration. Please revise the statement.

6. In Figure 4, D=directionality, but exactly how is directionality defined? Please be more specific.

[We asked Reviewer #4 to comment on the concerns of Reviewer #3, who was unable to submit a report. We relay these comments, which were submitted separately, below:

"I was asked to comment on the response to concerns raised by Reviewer #3. Please feel free to send these remarks to the authors.

In addition to my first comment to the authors, the other concern of Reviewer #3 is about the quantitative nature of SHG and THG imaging. In my opinion, the revised text on page 12 falls short. I would like to see a brief description of the types of quantitative information (even relative quantification) that can be obtained using SHG/THG imaging, citing for example the work of LaComb et al (2008) mentioned in the response letter.

Also, in response to Reviewer #1, the authors have produced very nice images of the lymph node. I suggest including it as a supplementary figure as it is useful information. Note that the differences in THG intensity and morphology (hollow core for lymphocytes etc) are consistent with those reported by Huang et al (Ref. 21) and this reference should be cited here. In addition, I suspect that T cells and B cells cannot be distinguished by SLAM microscopy. Perhaps the authors could comment on this as a limitation of the label free technique."]

REVIEWERS' COMMENTS:

Reviewer #1 (Remarks to the Author):

While the authors have had a reasonable go at addressing the comments I raised, it seems like an opportunity lost that the work they have done to address them has not been incorporated into the manuscript, where it could provide evidence supporting some of their claims.

In the Response Figure 1, the authors show some images of a lymph node showing likely lymphocytes and macrophages (the claimed neutrophils are less clear), illustrating to some extent the capacity of this system to differentiate between cell types.

In the Response Figure 2, the authors show some images showing combined label-free and anti-CD45-labelled images from a tumour. Unfortunately the two frames do not show the exact same region, so they don't allow an optimal comparison to be made. A more simple way forward would be to display the right hand panel twice, once with the anti-CD45 signal turned off, so that the label-free labelling of the anti-CD45-stained cells can be assessed more readily. However, if displayed appropriately, this would make an excellent addition to the manuscript.

It is a pity that these new pieces of data were not included in the manuscript at this revision, as particularly for Response Figure 2, it makes an important point. Indeed it is notable that in the revised supplementary Figures, Supplementary Figure 3 purports to show the same regions of tissue using label-free SLAM microscopy and histology. Using histology it is extremely difficult to be sure that you are visualising the same area of tissue you were examining under an intravital microscope. In contrast the approach attempted in Response Figure 2 allows visualisation of the identical field with and without label-free microscopy (simply by turning on and off the anti-CD45 channel), and is therefore much better than a histological approach, but it is not included in the manuscript! I suggest the authors work to find a way to include and describe particularly Response Figure 2 in the Supplementary data of revised version.

Response: We understand the reviewer's suggestion that Response Fig. 2 together with Supplementary Fig. 3 (H&E correlation) will make the manuscript stronger. Therefore, we included the modified Response Figure 2 as Supplementary Fig. 5 and its description as Supplementary Note 3 in the new manuscript.

"We injected a CD45-based marker (Monoclonal antibody OX1, eBioscience) into a living rat and acquired images of the tumor microenvironment before and after labelling (Supplementary Figure 5). Before labelling, the blue channel visualized mostly NADH-rich vesicles, while the yellow channel showed FAD-rich stroma cells and the purple channel picked up signals from THG-visible hollow spheres (6-8 μm in size), which are supposedly lymphocytes. After labelling,

stained leukocytes should appear in the blue channel while the assignment for the other channels remain the same. As shown in Supplementary Figure 5, there are three groups of cells in this dataset: 1) yellow stroma cells that remain yellow after staining, which validates the specificity of both THG and CD45 for leukocytes; 2) leukocytes with overlapping THG and CD45 signals, which confirms the identify of these THG-visible cells; 3) leukocytes with CD45 but no THG signals, which suggests that SLAM microscopy only visualizes subpopulations of leukocytes. This might be due to the fact that THG is not highly efficient for detecting all lymphocytes as they carry few lipid granules. It could also be caused by technical issues, such as a slight change of the imaging depth before and after staining due to the injection procedure.”

Supplementary Fig. 5. SLAM microscopy of the tumor microenvironment of a living rat before (a) and after (b,c) CD45 staining. The blue channel was assigned to NADH before staining and assigned to CD45 after staining. The images after staining (with or without CD45 channel) show cells with overlapping THG (magenta) and CD45 (blue) signals, demonstrating the specificity of THG signals. Scale bar: 50 μm .

Response Fig. 1, however, was used to answer Reviewer 1's question on whether we can also see lymphocytes using SLAM microscopy, which was an interesting question and experiment for us. However, we do not believe this figure (ex vivo imaging of a lymph node) is important or necessary to convey the main message of this paper—which is label-free intravital imaging of the tumor microenvironment. In addition, with the inclusion of Response Fig. 2, we provided more evidence validating the capacity of this system to visualize leukocytes. Therefore, we believe this figure is not necessary for this manuscript and the current arrangement will make the best presentation.

Reviewer #2 (Remarks to the Author):

In the revised version of the manuscript, the authors addressed all my concerns and significantly improved both the introduction and the discussion. I strongly recommend the manuscript for publication.

We thank this reviewer, and are pleased that we have not only address all of his/her concerns, but also that we have significantly improved our manuscript. We are pleased to hear that this reviewer strongly recommends publication.

Reviewer #4 (Remarks to the Author):

I thank the authors for carefully addressing many of the concerns raised in the previous review. There are however several remaining concerns:

1. I agree with Reviewer 3 that imaging leukocyte rolling, adhering and migrating are not new, and suggest placing less emphasis on this part of the paper. Specifically, the two sections headlined "Intravascular leukocyte recruitment" and "Interstitial leukocyte migration" can be combined and shortened significantly. Figures 3 and 4 can be moved to the supplement. Instead, more emphasis should be placed on the new results, namely the simultaneous capturing of cellular dynamics and metabolic profiles, to showcase the novel feature of SLAM microscopy. An expanded version of Supplementary Figure 6, with a more in depth analysis of the metabolic differences between migrating vs. cluster-forming leukocytes, should be moved to the main figure, accompanied by a new movie showing the tracks of the blue and the red cells so that readers have a better sense of how these cells moved.

Response: The main focus of this paper is to demonstrate the power of this technology (SLAM) in intravital imaging, especially in the context of the tumor microenvironment. We believe that Fig. 3 and Fig. 4 show exactly the intercellular dynamics we intended to show in this paper, and we feel that these are best presented in the main text of the manuscript.

Supplementary Fig. 6 should be sufficient to answer the reviewer's original question on the potential of SLAM microscopy to capture cellular dynamics together with their metabolic profiles. While we appreciate the interest and suggestion of this reviewer, we feel that the request of an expanded version of Supplementary Fig. 6 showing dynamic cell tracking through 2000+ frames is excessive for the scope of this paper, but would certainly be a focus for a more in depth focused study in the future.

2. In response to my previous question about ratiometric analysis, the authors propose to carry out "diligent calibration" and use "customized ratio metrics" (p.20 of the response letter). Please

described exactly how this is going to be done in the manuscript (not just in the letter) so that the analysis can be reproduced by other labs.

Response: All the redox analysis in this work can be reproduced by directly taking the value as $I_{2PF}/(I_{2PF} + I_{3PF})$, as shown on Page 11 in the main text. The proposed calibration and metrics for future work is now included in the main text as follows:

“Since 2PAF and 3PAF are simultaneously collected and expected to have identical incident power, we could potentially derive the relative concentration of NADH and FAD based on the known power dependence (power 2 for FAD and power 3 for NADH) and excitation efficiency of the system (calibration data from FAD and NADH solutions), which should provide a reliable qualitative assessment of the redox state.”

3. Regarding the question what triggered cluster formation, the author presented a reasonable argument that it is probably not due to laser damage (p.21 of the response letter). However, in response to a following concern about the sluggish blood flow, the authors pointed out that it is hard to avoid accidental vessel damage during the surgical procedure. Might the injury be the trigger for the leukocyte clustering? Again please address the issue of surgical damage in the manuscript and not just in the letter so the readers are aware of the potential confounding factors when attempting to reproduce the results.

Response: We have included these potential confounding factors in the main text.

“Leukocyte clustering is known to be triggered by various conditions of inflammation, including tumor progression, surgical operation, and laser damage. The possibility of this clustering being caused by the surgical procedure or laser damage is low, as the same experiments have been performed on control animals with no similar phenomena observed. Nevertheless, more controlled experiments are needed to pinpoint the exact trigger for this leukocyte cluster formation in future studies.”

4. Related to the point above, please restrict the use of the term "non-perturbative", which is used repeatedly in the manuscript.

Response: We'd like to emphasize that the term “non-perturbative” is often used to describe label-free multiphoton imaging for two reasons. First, compared to label-based imaging, no exogenous stains or tissue processing are introduced into the biological system prior to or during the imaging session. Second, compared to label-free technologies like mass spectroscopy imaging, tissue is not destroyed in the process, and can be repeatedly imaged. For example, in Fig. 4, no apparent damage was observed, and the tissue is still vital at the end of the video (after 70 minutes of continuous imaging). However, we understand the reviewer's concern about the surgical procedure, and we have reduced the use of the term “non-perturbative” in relevant parts of the manuscript.

5. Regarding the cell undergoing reverse transmigration (p.23 of the letter and p.11 of the manuscript), since it is difficult to tell if the cell was outside the blood vessel in the first frame, it is just as likely that the cell was initially arrested inside the blood vessel and subsequently detached and disappeared in the flow, without undergoing transmigration. Please revise the statement.

Response: We have deleted phrases suggesting reverse migration in the revised manuscript.

6. In Figure 4, D=directionality, but exactly how is directionality defined? Please be more specific.

Response: As explained on Page 17 of the manuscript, “directionality was defined as the displacement divided by the total path length of the cell.”

[We asked Reviewer #4 to comment on the concerns of Reviewer #3, who was unable to submit a report. We relay these comments, which were submitted separately, below:

"I was asked to comment on the response to concerns raised by Reviewer #3. Please feel free to send these remarks to the authors.

In addition to my first comment to the authors, the other concern of Reviewer #3 is about the quantitative nature of SHG and THG imaging. In my opinion, the revised text on page 12 falls short. I would like to see a brief description of the types of quantitative information (even relative quantification) that can be obtained using SHG/THG imaging, citing for example the work of LaComb et al (2008) mentioned in the response letter.

Response: We have included a brief description of the types of quantitative information that can be obtained using SHG/THG imaging, citing previous work.

“Although quantitative analysis of molecular structure based on SHG and THG intensity has been controversial due to their coherent nature^{12,48}, previous works have shown extensive analysis of the bulk optical parameters as well as morphological descriptions based on SHG and THG images^{12,49}.”

Also, in response to Reviewer #1, the authors have produced very nice images of the lymph node. I suggest including it as a supplementary figure as it is useful information. Note that the differences in THG intensity and morphology (hollow core for lymphocytes etc) are consistent with those reported by Huang et al (Ref. 21) and this reference should be cited here. In addition, I suspect that T cells and B cells cannot be distinguished by SLAM microscopy. Perhaps the authors could comment on this as a limitation of the label free technique."]

Response: Response Fig. 1 was used to answer Reviewer 1's question on whether we can also see lymphocytes using SLAM microscopy, which was an interesting question and experiment for us. However, we do not believe that this figure (ex vivo imaging of lymph node) is important or necessary to convey the main message of this paper—that being label-free intravital imaging of the tumor microenvironment. In addition, with the inclusion of Response Fig. 2, we provided more evidence validating the capacity of this system to visualize leukocytes. Therefore, we believe this figure is not necessary for this manuscript and the current arrangement will make the best presentation.